# Associations between moderate alcohol consumption, brain iron, and cognition in UK Biobank participants: Observational and mendelian randomization analyses

Anya Topiwala[1]*, Chaoyue Wang[2], Klaus P. Ebmeier[3], Stephen Burgess[4,5], Steven Bell[6], Daniel F. Levey[7,8], Hang Zhou[7,8], Celeste McCracken[9], Adriana Roca-Fernández[10], Steffen E. Petersen[11,12,13,14], Betty Raman[9], Masud Husain[2,15,16,17], Joel Gelernter[7,8], Karla L. Miller[2], Stephen M. Smith[2], Thomas E. Nichols[1,2]

1 Nuffield Department Population Health, Big Data Institute, University of Oxford, Oxford, United Kingdom, 2 Wellcome Centre for Integrative Neuroimaging (WIN FMRIB), Oxford University, Oxford, United Kingdom, 3 Department of Psychiatry, University of Oxford, Warneford Hospital, Oxford, United Kingdom, 4 MRC Biostatistics Unit, School of Clinical Medicine, University of Cambridge, Cambridge, United Kingdom, 5 Department of Public Health and Primary Care, School of Clinical Medicine, University of Cambridge, Cambridge, United Kingdom, 6 Department of Clinical Neurosciences, University of Cambridge, United Kingdom, 7 Department of Psychiatry, Yale University School of Medicine, New Haven, Connecticut, United States of America, 8 Department of Psychiatry, Veterans Affairs Connecticut Healthcare System, West Haven, Connecticut, United States of America, 9 Division of Cardiovascular Medicine, Radcliffe Department of Medicine, University of Oxford, National Institute for Health Research Oxford Biomedical Research Centre, Oxford University Hospitals NHS Foundation Trust, Oxford, United Kingdom, 10 Perspectum Diagnostics Ltd., Oxford, United Kingdom, 11 William Harvey Research Institute, NIHR Barts Biomedical Research Centre, Queen Mary University of London, Charterhouse Square, London, United Kingdom, 12 Barts Heart Centre, St Bartholomew's Hospital, Barts Health NHS Trust, West Smithfield, London, United Kingdom, 13 Health Data Research UK, London, United Kingdom, 14 Alan Turing Institute, London, United Kingdom, 15 Department of Experimental Psychology, University of Oxford, Oxford, United Kingdom, 16 Nuffield Department of Clinical Neuroscience, University of Oxford, Oxford, United Kingdom, 17 Division of Clinical Neurology, John Radcliffe Hospital, Oxford University Hospitals Trust, Oxford, United Kingdom

* anya.topiwala@bdi.ox.ac.uk

**Data Availability Statement:** Imaging and observational data underlying the results presented

## Abstract

### Background

Brain iron deposition has been linked to several neurodegenerative conditions and reported in alcohol dependence. Whether iron accumulation occurs in moderate drinkers is unknown. Our objectives were to investigate evidence in support of causal relationships between alcohol consumption and brain iron levels and to examine whether higher brain iron represents a potential pathway to alcohol-related cognitive deficits.

### Methods and findings

Observational associations between brain iron markers and alcohol consumption ($n$ = 20,729 UK Biobank participants) were compared with associations with genetically predicted alcohol intake and alcohol use disorder from 2-sample mendelian randomization (MR). Alcohol intake was self-reported via a touchscreen questionnaire at baseline (2006 to 2010). Participants with complete data were included. Multiorgan susceptibility-weighted

are available from the UK Biobank upon successful application (https://www.ukbiobank.ac.uk/enable-yourresearch/apply-for-access). Genetic summary statistics for serum iron measures are freely available (https://www.decode.com/summarydata/), as are GSCAN summary statistics (https://genome.psych.umn.edu/index.php/GSCAN). Summary statistics for alcohol use disorder are available upon application through dfGaP at accession no. phs0016732.v3.p1 (https://www.ncbi.nlm.nih.gov/projects/gap/cgi-bin/study.cgi study_id=phs001672.v3.p1).

**Funding:** AT is supported by a Wellcome Trust (https://wellcome.org/) fellowship (216462/Z/19/Z). CW is funded, in part, by the China Scholarship Council (CSC, https://www.chinesescholarshipcouncil.com/). KPE is funded by the UK Medical Research Council (https://mrc.ukri.org/, G1001354 & MR/K013351/1) and the European Commission (https://ec.europa.eu/programmes/horizon2020/en/home, Horizon 2020 732592). CM is funded by the NIHR Oxford Biomedical Research Centre (IS-BRC-1215-20008) and the BHF Centre of Research Excellence, Oxford. JG, DL and HZ are supported by the US Department of Veterans Affairs (https://www.research.va.gov/funding/, I01CX001849). SBu is supported by a Sir Henry Dale Fellowship jointly funded by the Wellcome Trust and the Royal Society (https://royalsociety.org/, 204623/Z/16/Z). SBe was supported by the British Heart Foundation (https://www.bhf.org.uk/for-professionals/information-for-researchers/what-we-fund, RG/16/4/32218) and the NIHR Cambridge Biomedical Research Centre (BRC-1215-20014). MH is supported by the Wellcome Trust (206330/Z/17/Z) and NIHR Oxford Biomedical Research Centre (IS-BRC-1215-20008). SS is supported by a Wellcome Trust Collaborative Award 215573/Z/19/Z. KLM is supported by a Wellcome Trust Senior Research Fellowship (202788/Z/16/Z). TN is supported by the Li Ka Shing Centre for Health Information and Discovery, an NIH grant (https://www.nih.gov/, TN: R01EB026859), the National Institute for Health Research Oxford Biomedical Research Centre (BRC-1215-20014), and a Wellcome Trust award (TN: 100309/Z/12/Z). The funders had no role in study design, data collection and analysis, decision to publish, or preparation of the manuscript.

**Competing interests:** I have read the journal's policy and these authors have the following competing interests: BR has consulted for Axcella Therapeutics in the past. AR is employed by Perspectum Ltd. SBu is a paid statistical consultant on PLOS Medicine's statistical board. JG is named as an inventor on PCT patent application #15/

magnetic resonance imaging (9.60 ± 1.10 years after baseline) was used to ascertain iron content of each brain region (quantitative susceptibility mapping (QSM) and T2*) and liver tissues (T2*), a marker of systemic iron. Main outcomes were susceptibility ($\chi$) and T2*, measures used as indices of iron deposition. Brain regions of interest included putamen, caudate, hippocampi, thalami, and substantia nigra. Potential pathways to alcohol-related iron brain accumulation through elevated systemic iron stores (liver) were explored in causal mediation analysis. Cognition was assessed at the scan and in online follow-up (5.82 ± 0.86 years after baseline). Executive function was assessed with the trail-making test, fluid intelligence with puzzle tasks, and reaction time by a task based on the "Snap" card game.

Mean age was 54.8 ± 7.4 years and 48.6% were female. Weekly alcohol consumption was 17.7 ± 15.9 units and never drinkers comprised 2.7% of the sample. Alcohol consumption was associated with markers of higher iron ($\chi$) in putamen (β = 0.08 standard deviation (SD) [95% confidence interval (CI) 0.06 to 0.09], $p < 0.001$), caudate (β = 0.05 [0.04 to 0.07], $p < 0.001$), and substantia nigra (β = 0.03 [0.02 to 0.05], $p < 0.001$) and lower iron in the thalami (β = −0.06 [−0.07 to −0.04], $p < 0.001$). Quintile-based analyses found these associations in those consuming >7 units (56 g) alcohol weekly. MR analyses provided weak evidence these relationships are causal. Genetically predicted alcoholic drinks weekly positively associated with putamen and hippocampus susceptibility; however, these associations did not survive multiple testing corrections. Weak evidence for a causal relationship between genetically predicted alcohol use disorder and higher putamen susceptibility was observed; however, this was not robust to multiple comparisons correction. Genetically predicted alcohol use disorder was associated with serum iron and transferrin saturation. Elevated liver iron was observed at just >11 units (88 g) alcohol weekly c.f. <7 units (56 g). Systemic iron levels partially mediated associations of alcohol intake with brain iron. Markers of higher basal ganglia iron associated with slower executive function, lower fluid intelligence, and slower reaction times. The main limitations of the study include that $\chi$ and T2* can reflect changes in myelin as well as iron, alcohol use was self-reported, and MR estimates can be influenced by genetic pleiotropy.

## Conclusions

To the best of our knowledge, this study represents the largest investigation of moderate alcohol consumption and iron homeostasis to date. Alcohol consumption above 7 units weekly associated with higher brain iron. Iron accumulation represents a potential mechanism for alcohol-related cognitive decline.

## Author summary

### Why was this study done

- There is growing evidence that even moderate alcohol consumption negatively impacts the brain, but the mechanisms underlying this are unclear.

878,640 entitled: "Genotype-guided dosing of opioid agonists," filed January 24, 2018 and issued on January 26, 2021 as U.S. Patent No. 10,900,082. TN "Paid statistical consultancy, Perspectum". The other authors declare no competing financial interests.

**Abbreviations:** AUD, alcohol use disorder; BMI, body mass index; CMA, causal mediation analysis; CI, confidence interval; cT1, corrected T1; FDR, false discovery rate; GWAS, genome-wide association study; IDP, image-derived phenotype; IVW, inverse variance weighted; MR, mendelian randomization; PDFF, proton density fat fraction; QSM, quantitative susceptibility mapping; SD, standard deviation; SNP, single nucleotide polymorphism; swMRI, susceptibility-weighted MRI; TMT, trail-making test; UKB, UK Biobank.

- One possibility is that accumulation of iron in the brain could contribute, as higher brain iron has been described in numerous neurodegenerative conditions including Alzheimer's and Parkinson's disease.

- To the best of our knowledge, there have been no studies investigating if brain iron levels differ by level of alcohol consumption.

### What did the researchers do and find

- In 20,965 participants in a United Kingdom cohort study, we explored relationships between self-reported alcohol consumption and brain iron levels, measured using magnetic resonance imaging.

- We assessed the association of alcohol intake with blood and liver iron and cognitive measures associated with higher brain iron.

- Alcohol consumption above 7 units (56 g) weekly was associated with markers of higher iron in the basal ganglia, which in turn associated with worse cognitive function.

- These observational findings were further supported by analyses using genetic variants as proxies for alcohol consumption.

### What do these findings mean

- These findings suggest that moderate alcohol consumption is associated with higher iron levels in the brain.

- Brain iron accumulation represents a potential mechanism for alcohol-related cognitive decline.

- Key limitations are that changes in myelin may also alter imaging markers and alcohol intake was self-reported. It is unclear how our findings generalize to other populations, particularly those which are more ethnically diverse and socioeconomically deprived.

### Introduction

There is growing evidence that moderate alcohol consumption adversely impacts brain health, contradicting earlier claims [1,2]. Given the high prevalence of moderate drinking, even small causal associations have substantial population impact [3]. Clarity about the pathological mechanisms by which alcohol acts upon the brain is vital not just for disease aetiology, but also to offer opportunities for intervention.

One largely neglected possibility is that iron overload contributes to alcohol-related neurodegeneration [4,5]. While neurological sequelae of inherited iron overload disorders have long been recognised [6], higher brain iron has now been implicated in the pathophysiology of Alzheimer's and Parkinson's diseases [7,8]. Intriguingly, not only does the clinical profile of

alcohol-related dementia overlap with these disorders, but also recent observational evidence suggests heavy alcohol use may associate with iron accumulation in the brain [9,10].

What has not previously been explored is whether brain iron accumulation is observed with moderate alcohol consumption, and if so, whether these associations are causal. Furthermore, the mechanisms by which alcohol could influence brain iron and whether there are clinical consequences of subtle elevations in brain iron are unknown. Low levels of drinking have been observationally associated with blood markers of iron homeostasis. However, studies to date have been small, have neglected genetic contributions to iron accumulation (polymorphisms predicting serum iron are highly prevalent in European populations [11]), and serum markers may be poorly specific for body iron stores [12–14]. Better insights require well-powered samples with genetic data and concurrent measurement of iron accumulation in brain and liver, the most reliable indicator of the body's iron stores [15,16]. If iron deposition is mechanistically involved in alcohol's effect on the brain, there are potential opportunities for earlier monitoring via serum iron markers, as well as intervention with chelating agents [17,18].

In this study, to the best of our knowledge, we performed the largest multiorgan investigation into alcohol-related iron homeostasis to date. Alcohol consumption weekly was divided into quintiles, and never and previous drinkers distinguished. Conventional observational and genetic analyses were triangulated to investigate causal effects (Fig 1). Our objective was to characterise the dose–response relationship of alcohol consumption and brain iron, in observational and mendelian randomization (MR) frameworks. MR is a method that, under specific

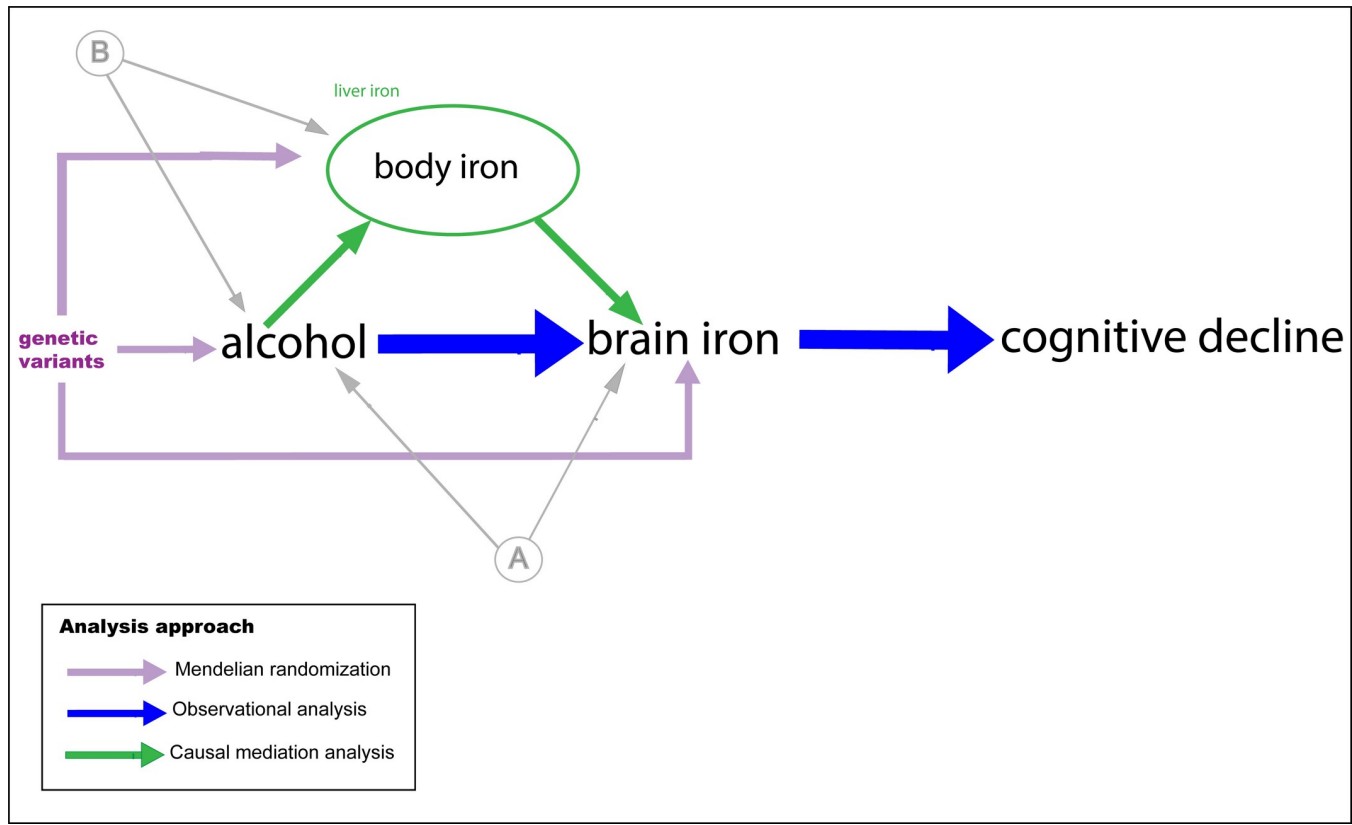

**Fig 1. Hypothesised model and analysis approach.** (A) and (B) represent unmeasured confounding. Brain iron is measured using T2* and susceptibility (UKB), and body iron stores proxied using liver T2* (UKB). UKB, UK Biobank.

assumptions, seeks to estimate causal effects. Furthermore, we sought to investigate whether alcohol influences brain iron via changes in systemic iron levels. Lastly, we explored whether higher brain iron represents a potential pathway to alcohol-related cognitive deficits.

## Methods

### Participants

Participants were scanned as part of the UK Biobank (UKB) study [19], which recruited volunteers aged 40 to 69 years in 2006 to 2010. Invitations for imaging were sent to all participants. Interested individuals then underwent screening to assess if they were safe and able to tolerate imaging. To date, approximately 50,000 participants have brain scans analysed and approximately 15,000 have abdominal imaging. UKB received ethical approval from the Research Ethics Committee (reference 11/NW/0382), and all participants provided written informed consent. All participants with complete data were included (S1 Fig). No exclusions were made on the basis of dementia diagnosis as no participants with complete data had dementia at the time of imaging. As our study was conducted using existing resources to test an a priori hypothesis, we did not publish a prespecified analysis plan before conducting analyses between November 2021 and January 2022. Further analyses were subsequently performed in March 2022 in response to peer review where highlighted. This study is reported as per the Strengthening the Reporting of Observational Studies in Epidemiology (STROBE) guideline, including guidelines specific for MR studies (S1 and S2 Checklists).

### Alcohol consumption

Alcohol intake was self-reported at study baseline through a touchscreen questionnaire. Participants identified themselves as either current, never, or previous drinkers. All groups were included if they had complete data on alcohol intake. For current drinkers, total weekly number of United Kingdom units (1 unit = approximately 8 g; a United States standard drink is 14 g) of alcohol consumed was calculated by summing across beverage types as previously described [20]. To directly compare associations with brain iron at different levels of alcohol intakes, weekly consumption was categorized into quintiles (and octiles for a sensitivity analysis) in current drinkers. The lowest quintile of drinkers was used as the reference category to avoid underestimating alcohol-related risks [21].

### Brain imaging

Participants underwent brain MRI at 3 imaging centres (Newcastle upon Tyne, Stockport, or Reading) with identical Siemens Skyra 3T scanners (software VD13) using a standard 32-channel head coil (release date 02.02.2021). Susceptibility-weighted MRI (swMRI) data were used for this study (3D GRE, TE1/TE2/TR = 9.4/20/27 ms, voxel size = $0.8 \times 0.8 \times 2.0$ mm) as a measure sensitive to magnetic tissue constituents. Detailed image preprocessing and quality control pipelines are described elsewhere [22]. Brain iron content was ascertained using quantitative susceptibility mapping (QSM) and $T2^*$, both derived from swMRI data. $T2^*$ reflects differences in tissue microstructure related to iron (sequestered to ferritin) and myelin and correlates with postmortem estimates of iron deposits in brain grey matter [23]. Susceptibility reflects the net (sequestered and non-sequestered) content of susceptibility-shifting sources like iron and myelin. Two distinct and complementary metrics of brain iron deposition were used, $T2^*$ and QSM, to produce image-derived phenotypes (IDPs). While these metrics are coupled, consistent findings across the 2 will provide greater evidence that iron levels are affected. Subject-specific masks for 14 subcortical regions were derived from the T1-weighted

structural brain scan. We then calculated IDPs corresponding to the median $T2^*$ and $\chi$ values for each region. The 14 regions correspond to left and right of the 7 subcortical structure regions of interest obtained from the T1 image: putamen, caudate, hippocampus, amygdala, pallidum, thalamus, and accumbens. Two additional IDPs were calculated from QSM (left and right substantia nigra) that were not available for $T2^*$. In brief, $T2^*$ values were calculated from magnitude data. $T2^*$-induced signal decay was calculated from the 2 echo times. $T2^*$ images were spatially filtered to reduce noise and transformed into the T1 space (by linear registration). QSM depends on phase images, which were obtained from individual coil channels, combined, masked, and unwrapped. Magnetic susceptibility ($\chi$) was calculated using a QSM pipeline including background field removal, dipole inversion, and CSF referencing as described elsewhere [10]. Median $\chi$ values (in parts per billion) across voxels within each region were calculated, resulting in 16 QSM IDPs.

## Liver imaging

Systemic iron levels were estimated using MRI-derived liver iron (a marker of liver iron levels [24]). During the same visit that brain imaging was performed, participants also underwent abdominal imaging on a Siemens 1.5T Magnetom Aera [25]. The acquisition used the Liver-MultiScan protocol from Perspectum Diagnostics (thus the data for liver MRI indices is available for a fewer number of participants than brain MRI measures to date). A multi-echo spoiled gradient-echo single breath hold MRI sequence was acquired as a single transverse slice through the centre of the liver. Three ROI, representative of the liver parenchyma, were selected and mean $T2^*$ calculated. Iron levels were converted to mg/g. One IDP is liver fat, indexed using proton density fat fraction (PDFF, %), validated against MR spectroscopy and liver biopsy, and has a high specificity and sensitivity for non-alcoholic fatty liver disease [26,27]. The other IDP, corrected T1 (cT1, milliseconds) was used as a marker of inflammation and fibrosis [28]. The data for liver MRI indices are available for a fewer number of participants than brain MRI measures to date.

## Genetic variants

Genetic instruments for alcohol, susceptibility, and serum iron markers were selected based on the sentinel variants at genome-wide significant loci ($p < 5 \times 10^{-8}$) reported in the largest publicly available European ancestry genome-wide association studies (GWAS) (**S1 Table**). The distinct genetic architecture between different alcohol use traits motivates their separate analysis. Genetically predicted alcohol consumption was instrumented using 91/92 (depending on single nucleotide polymorphism (SNP) or suitable proxy availability in outcome data) independent SNPs. These variants were associated with alcohol consumption (log-transformed drinks per week) in the largest published GWAS comprising 941,280 individuals (GWAS and Sequencing Consortium of Alcohol and Nicotine) [29]. Genetic associations were used from the sample excluding UKB to avoid bias towards the observational estimate for overlapping samples [30]. Alcohol use disorder (AUD) was instrumented using 24 conditionally independent genome-wide significant genetic variants in the largest published GWAS comprising the Million Veterans Program and the Psychiatric Genomics Consortium [31]. AUD cases were defined using ICD 9/10 codes within the MVP ($n = 45,995$) and DSM-IV within the PGC ($n = 11,569$). Genetic associations with serum markers of iron homeostasis (serum iron, ferritin, transferrin saturation (of iron-binding sites of transferrin occupied by iron), and total iron-binding capacity) were obtained from the largest published GWAS to date (meta-analysis of deCODE, INTERVAL, and Danish Blood Donor Study) [32]. Genetic associations with brain $T2^*$ and susceptibility (IV-outcomes) were obtained from the largest GWAS of brain

imaging phenotypes [10,33]. All genetic associations were based on GWAS of European ancestry samples. A summary of cohorts comprising the relevant GWAS is given in **S1 Table**. All publicly available deidentified summary data used have ethical permissions from their respective institutional review boards.

## Clinical measures

We utilized the expanded cognitive battery performed on the subset of participants who underwent imaging, which included: trail-making tests (TMTs) (durations, reflecting executive function; numerical–"TMT A;" alphanumeric–"TMT B"), prospective memory (incorrect or correct on first/second attempt on a shape task on screen), and fluid intelligence (sum of correct answers on questions designed to assess logic and reasoning) [34]. A larger subset of the wider UKB study (approximately 100 k) was invited by email to undertake further web-based questionnaires at online follow up (mean 5.82 ± 0.86 years after study baseline). These included: TMT, fluid intelligence, digit span (maximum digits recalled, reflecting working memory), pairs matching (number correctly associated, reflects visual memory), and symbol digit substitution (use of a code to substitute symbols for digits). At baseline only fluid intelligence, prospective memory, digit span, and pairs matching were administered. Motor function was assessed using simple reaction speed in the cognitive battery (mean time to correctly identify matches in a task based on the "Snap" card game), handgrip strength (measured using a hydraulic hand dynamometer), and self-reported gait speed (slow/steady/brisk) at baseline.

## Statistical analysis

All analyses were performed in R (version 3.6.0).

## Observational

Separate linear regression models were used to assess the relationship between (1) alcohol consumption and $\chi$/T2$^*$; (2) alcohol consumption and liver iron; and (3) susceptibility and neurocognitive measures (ordered logistic regression for gait speed). Variables were quantile normalized to enforce Gaussianity in the data. Covariates previously associated with alcohol intake or iron levels and image-related confounders were included in the models. Age in years, sex, smoking status (reported as categorical variable: never/previous/current), body mass index (BMI) (calculated from measured height and weight), blood pressure (automated measurement), and cholesterol (from blood sample) were assessed at baseline. As heavy alcohol can impact blood pressure and cholesterol, models were also run excluding these variables to check we were not including non-confounders in our models [35]. Educational qualifications were self-reported in categories, as was household income. Historical job type was coded according to the Standard Occupational Classification 2000 [36], and Townsend Deprivation Index used as a continuous measure of deprivation based on census information. Diabetes was coded as a binary variable (presence of an ICD code for non-insulin dependent, insulin dependent, or nonspecified diabetes, E10, 11&14). The full set of imaging confounds was used as recently proposed [37] which include imaging site (3 sites), head motion, and head position.

For the subset with no missing data, dietary factors (meat, fish, bread, fruit, and vegetable consumption) and dietary iron supplementation were also included as covariates to test whether differences in diets according to alcohol intake were driving associations. Diet and supplement data were assessed at baseline via the questionnaire.

Analyses were also controlled for genetic polymorphisms prevalent in European populations that affect iron absorption and metabolism and have been linked to brain phenotypes [38]. A risk score was calculated by multiplying the dosage of 3 SNPs by their associations with

serum iron (betas from GWAS [39]): rs1800562—in the HFE gene that is the main cause of hereditary hemochromatosis and carried by approximately 5% to 10% of Europeans [40]; rs1799945—also in HFE, carried by approximately 15% Europeans; and rs855791—which modulates hepicidin and approximately 53% Europeans are heterozygotes. No subjects were excluded on the basis of iron chelator prescription as none were documented. In a sensitivity analysis of females, menopause status at imaging was added as a covariate in an analysis of the brain region where the strongest observational associations were observed (putamen), as menstruation is protective against phenotypic expression of haemochromatosis [41]. In response to peer review comments, interactions between alcohol and age and sex were explored, as well as quadratic terms for alcohol to examine for nonlinear effects of alcohol.

For models with cognitive measures as the dependent variable, which are age-dependent, age interactions were included to explore whether iron altered age-related decline. Separate models were used to test cognitive performance at each time point. For models with $\chi$/T2* as the dependent variable, an interaction between alcohol and liver iron was tested to assess for synergistic neurotoxic effects of alcohol and iron. To adjust for multiple testing, Bonferroni and false discovery rate (FDR, 5%) corrected $p$ values were calculated. Correction methods were separately applied to models testing susceptibility-alcohol associations and those testing susceptibility-cognitive associations.

Causal mediation analysis (CMA) was performed to understand the mechanisms by which alcohol impacts brain iron [42,43]. CMA, unlike traditional mediation methods, defines, identifies, and estimates causal mediation effects without reference to a specific statistical model. Liver iron, reflecting systemic iron levels, was used as a mediator. The liver is the body's primary iron store, and liver iron has better specificity for systemic iron stores than serum ferritin, which is an acute phase reactant liable to fluctuate according to inflammatory processes [15]. CMA tests the statistical significance of the direct from indirect (mediated) effects of alcohol on brain iron. CMA was run on participants with complete data. Two separate multiple linear regression models were run. First, with putamen susceptibility as the dependent variable, with alcohol, age, sex, Townsend Deprivation Index, income, qualifications, historical job code, diabetes, smoking, blood pressure, iron genetic risk score, BMI, cholesterol, and liver iron as covariates. In the second model, the mediator, liver T2*, was the dependent variable, with all covariates as in the first model. CMA was then run using nonparametric bootstrapping to generate confidence intervals (CIs) (1,000 simulations). Assumptions in CMA include no unmeasured confounding between the exposure and outcome, between the exposure and mediator, or between the mediator and the outcome. In response to peer review, the mediation analysis was extended to include all brain regions.

Two sets of post hoc analyses were performed. First, given the prominence of basal ganglia iron levels in relation to alcohol we observed, associations with available motor phenotypes (simple reaction speed, handgrip strength, and self-reported walking pace) were sought. Second, given observed associations between brain iron measures and executive function, we assessed associations between alcohol consumption and executive function (separately at time of scan and online follow up). In response to peer review, quadratic terms for alcohol were examined to test for nonlinear behaviour in the effects of alcohol consumption on cognitive function. We could not examine relationships with relevant diseases, for example, Parkinson's due to insufficient numbers within the imaging sub study.

## Mendelian randomization

MR aims to estimate causal relationships using observational data [44]. There are 3 key assumptions: (1) genetic variants are robustly associated with the exposure; (2) they share no common

cause with the outcome; and (3) that genetic variants only affect the outcome through the exposure. Two sample linear MR was used to obtain estimates for the association between genetically predicted alcohol consumption/AUD and (1) brain susceptibility; (2) serum markers of iron homeostasis (as part of our investigation into the pathway by which alcohol influences brain iron). Analyses were conducted using *MendelianRandomization* (version 0.5.1) and *TwoSampleMR* (version 0.5.6) R packages. Variant harmonization ensured the association between SNPs and exposure and that between SNPs and the outcome reflected the same allele. Palindromic variants, where harmonization could not be confirmed, were excluded. Strands were aligned between studies. No proxies were used given the availability of SNPs across datasets. Several robust MR methods were performed to evaluate the consistency of the causal inference. Inverse variance weighted (IVW) analysis (multiplicative random effects) regresses the effect sizes of the variant-iron marker associations against the effect sizes of the variant-alcohol associations. The MR-Egger method uses a weighted regression with an unconstrained intercept to relax the assumption that all genetic variants are valid IVs (under the Instrument Strength Independent of Direct Effect (InSIDE) assumption) [45]. A nonzero intercept term can be interpreted as evidence of directional pleiotropy, where an instrument is independently associated with the outcome violating an MR assumption. The median and modal MR methods (reported in supplementary figures) are also more resistant to pleiotropy, as they are robust when up to 50% of genetic variants or more than not, respectively, are invalid. These methods are recommended in practice for sensitivity analyses as they require different assumptions to be satisfied, and therefore if estimates from such methods are similar, then any causal claim inferred is more credible. Analyses performed on one of the 3 cohorts (INTERVAL) meta-analysed in the serum iron GWAS was adjusted for alcohol consumption, so to check whether anti-conservative bias was impacting associations, a sensitivity analysis was run using weights derived solely from deCODE summary statistics that were unadjusted for alcohol. To adjust for multiple testing, Bonferroni and FDR (5%) corrected *p* values were calculated. Power calculations for MR analyses were based on an online calculator developed by one of the authors [46].

## Results

Baseline characteristics of the included sample with complete data are shown in **Table 1**. Mean alcohol consumption was higher than current UK low risk guidelines (<14 units weekly), although within guidelines at the time for men (>21 units weekly pre-2016). Characteristics according to alcohol consumption are shown in **S2 Table**. Never drinkers comprised a higher proportion of females, lower blood pressure, and higher prevalence of diabetes compared to higher intake alcohol consumers.

### Associations with brain iron

**Observational analyses.** Alcohol consumption was associated with higher susceptibility in bilateral putamen (beta = 0.08, 95% CI: 0.06 to 0.09, *p* < 0.001), caudate (beta = 0.06, 95% CI: 0.04 to 0.06, *p* < 0.001), and substantia nigra (beta = 0.03, 95% CI: 0.02 to 0.05, *p* < 0.001) (**Fig 2** and **S3 Table**). Alcohol was associated with lower iron in the thalamus (beta = −0.06, 95% CI: −0.07 to −0.04, *p* < 0.001). Drinking greater than 7 units weekly was associated with higher susceptibility for all brain regions, except the thalamus (**Fig 3** and **S4 Table**). A sensitivity analysis with finer categorisation of alcohol intake revealed that associations were not observed at lower intakes than 7 units (compared to reference of <4 units weekly) (**S5 Table**). Menopause status did not associate with susceptibility in most brain regions in females, with the exception of slightly higher susceptibility in left thalamus (beta = 0.16, 95% CI: 0.04 to 0.27, *p* = 0.01) and left hippocampus (beta = 0.17, 95% CI: 0.05 to 0.29, *p* = 0.006) in females who

**Table 1. Baseline characteristics of UKB samples.** Alcohol is measured in UK units weekly. 1 unit = 8 grams ethanol. Educational qualifications were determined by self-report at baseline.

| | Sample with brain imaging N = 20,729 | Sample with liver and brain imaging N = 6,936 | Wider UKB sample N = 502,489[1] |
|---|---|---|---|
| **Age**[2], *years* | 54.8 ± 7.4 | 54.6 ± 7.3 | 56.53 ± 8.1 |
| **Sex,** *females* **n (%)** | 10,821 (48.6) | 3,556 (51.3) | 273,375 (54.4) |
| **Townsend Deprivation Index** | −2.0 ± 2.6 | −2.1 ± 2.6 | −1.3 ± 3.1 |
| **Alcohol**[2], *units weekly* | 17.7 ± 15.9 | 17.4 ± 15.6 | 17.1 ± 18.0 |
| **Current smoker, n (%)** | 1,390 (6.2) | 444 (6.4) | 52,977 (10.6) |
| **Education,** *college or university degree,* **n (%)** | 10,041 (48.6) | 3,452 (49.8) | 161,158 (32.7) |
| *A levels* | 2,732 (13.2) | 967 (13.9) | 55,321 (11.2) |
| *No qualifications* | 1,107 (5.3) | 341 (4.9) | 85,269 (17.3) |
| **BMI index**[2], *kg/m²* | 26.5 ± 4.0 | 26.1 ± 3.9 | 27.4 ± 4.8 |
| **Systolic blood pressure**[2], *mm Hg* | 136.9 ± 18.6 | 136.8 ± 18.7 | 139.7 ± 19.7 |
| **Diastolic blood pressure,** *mm Hg* | 81.6 ± 10.5 | 81.4 ± 10.5 | 82.2 ± 10.7 |

[1]Total UKB sample. N varies according to missing data as mean/% was calculated on maximal sample size for each variable.

[2]Mean ± standard deviation.

BMI, body mass index; UKB, UK Biobank.

had hysterectomies compared to premenopausal females (**S6 Table**). Controlling for menopause status did not alter associations between alcohol and susceptibility for any brain region (**S6 Table**), neither did excluding blood pressure and cholesterol as covariates (**S3 Table**). Inverse associations between thalamus susceptibility and alcohol were only observed in the heaviest drinking groups (**Fig 3**). There were significant interactions with age in bilateral putamen and caudate, but not with sex, smoking, or Townsend Deprivation Index (**S3 Table**). Quadratic terms for alcohol were not significant (**S3 Table**). In the subset with complete data (**S7** and **S8 Tables),** additional adjustment for dietary factors, including frequency of red meat consumption, and dietary iron supplementation did not change the pattern of associations. Findings with T2* were broadly consistent, although associations with alcohol were only observed in putamen and caudate (**S3 Table**). These are 2 regions where correlation between T2* and susceptibility measures are more highly correlated (r = −0.58 to 0.75).

**Genetic analyses.** Using 92 SNPs significantly associated with alcohol consumption, we found a positive association with left putamen susceptibility (IVW β = 0.25, 95% CI: 0.01 to 0.49, *p* = 0.04) and right hippocampus susceptibility (IVW β = 0.28, 95% CI: 0.05 to 0.50, *p* = 0.02) (**Fig 4**). However, neither passed FDR correction for multiple comparisons and there was evidence of heterogeneity between estimates (right hippocampus: Cochrane's Q statistic = 121.4, df = 91, *p* = 0.018; Cochrane's Q statistic = 137.4, df = 91, *p* = 0.001). Although alternative methods gave consistent estimates, 95% CIs were wide (**S2 Fig**). Genetically predicted AUD, instrumented with 24 SNPs, associated with higher right putamen susceptibility, although this did not survive multiple comparison correction (IVW β = 0.18, 95% CI: 0.001 to 0.35, *p* = 0.04) (**S3 Fig**). There were no other significant associations between genetically predicted AUD and other susceptibility measures. Based on a sample size of 29,579, $R^2$ = 0.003 and α = 0.05, the MR analysis had 65.4% power to detect a causal effect of 0.25 SD.

## Pathways from alcohol to brain iron

Higher systemic iron levels (liver iron) were found to partially mediate alcohol's association with bilateral putamen and caudate susceptibility in CMA (**Fig 5 and S9 Table**). For example,

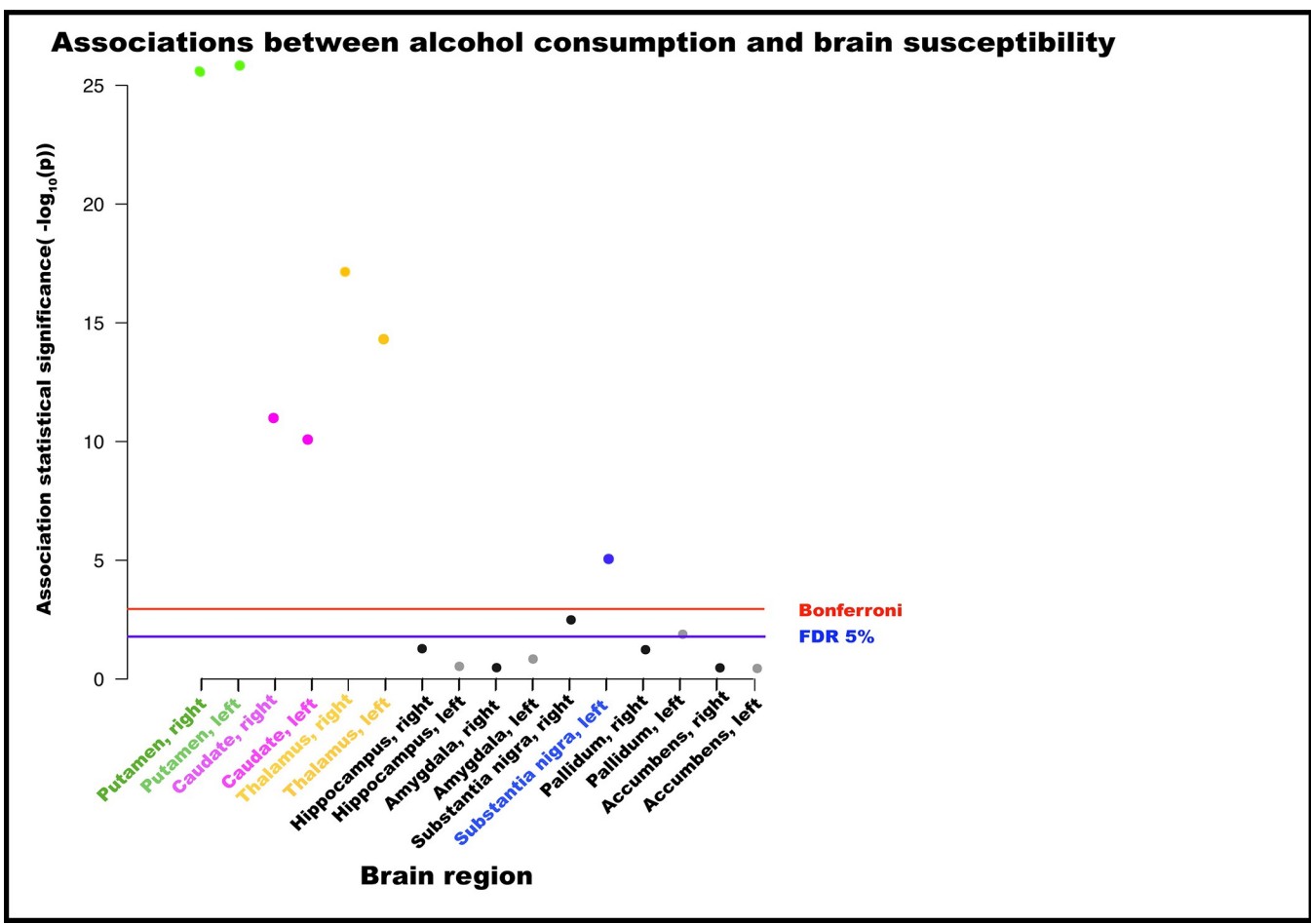

**Fig 2. Associations between alcohol consumption and QSM phenotypes.** Associations surviving multiple testing comparisons are coloured, with colours corresponding to their region labels for ease of viewing. Estimates represent SDs and were generated from regression models adjusted for: age, sex, smoking, BMI, educational qualifications, Townsend Deprivation Index, household income, historical job code, diabetes, cholesterol, blood pressure, rs1800562, rs1799945, rs855791, and full set of image-related confounds. BMI, body mass index; FDR, false discovery rate; QSM, quantitative susceptibility mapping; SD, standard deviation.

a 1 SD increase in weekly alcohol consumption was associated with a 0.05 (95% CI: 0.02 to 0.07, $p < 0.001$) mg/g increase in liver iron (**Fig 5**). Approximately 1 mg/g increase in liver iron was associated with a 0.44 (95% CI: 0.35 to 0.52, $p < 0.001$) SD increase in left putamen susceptibility. In this sample, 32% (95% CI: 22 to 49, $p < 0.001$) of alcohol's total effect on left putamen susceptibility was mediated via higher systemic iron levels (the indirect effect). The other 68% is mediated via other pathways (the direct effect). The proportion mediated was highest for the left caudate (48%, 95% CI: 0.30 to 0.94, $p < 0.001$) out of all brain regions. In contrast, we observed no significant mediation (after multiple testing correction) in the hippocampi, pallidum, accumbens, or substantia nigra (**S9 Table**).

Alcohol consumption greater than 11 units weekly was associated with higher liver iron measured by MRI, a robust marker of systemic iron stores in males (β = 0.05, 95% CI: 0.01 to 0.08, $p = 0.006$) (**S4 Fig**). In females, higher liver iron was observed at intakes greater than 17 units weekly (β = 0.06, 95% CI: 0.03 to 0.08, $p < 0.001$). This is within currently defined UK "low risk" guidelines (<14 units weekly). Iron appeared to be a more sensitive liver marker of alcohol consumption than fat (PDFF) or fibrosis (cT1) (**S5** and **S6 Figs**).

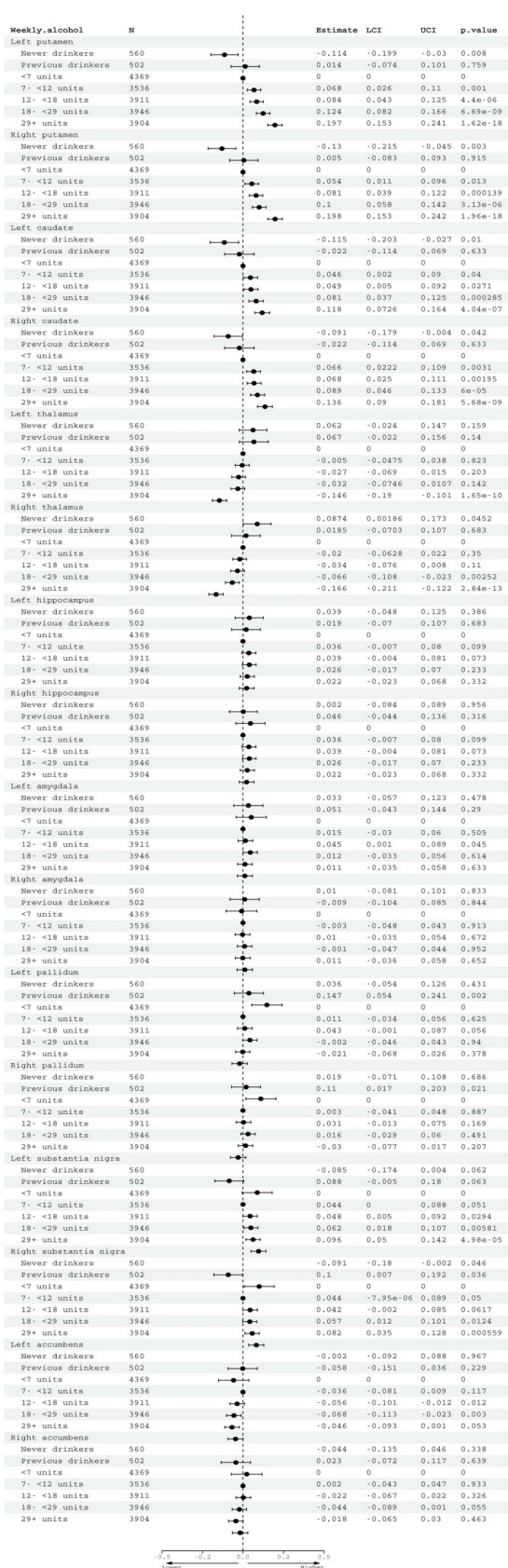

**Fig 3. Associations between alcohol consumption (quintiles) and QSM phenotypes.** Reference group is those consuming less than 7 units of alcohol weekly. Estimates generated from regression models adjusted for: age, sex, smoking, BMI, educational qualifications, Townsend Deprivation Index, household income, historical job code, diabetes, cholesterol, blood pressure, rs1800562, rs1799945, rs855791, and full set of image-related confounds. BMI, body mass index; LCI, lower confidence interval; N, number; QSM, quantitative susceptibility mapping; UCI, upper confidence interval.

Using 24 SNPs significantly associated with AUD, there were statistically significant associations between genetically predicted AUD and serum iron (IVW β = 0.12, 95% CI: 0.05 to 0.19, $p$ = 0.001) as well as transferrin saturation (IVW β = 0.11, 95% CI: 0.03 to 0.12, $p$ = 0.006) that survive even stringent Bonferroni multiple testing correction (**Fig 6**). There was no evidence of heterogeneity using Cochrane's Q (22.0, df = 23, $p$ = 0.52). CIs for MR-Egger estimates were larger (serum iron β = 0.17, 95% CI: 0.009 to 0.34; transferrin saturation β = 0.16, 95% CI: −0.02 to 0.33) but broadly consistent (**S7 Fig**). Although estimates for associations between genetically predicted alcohol consumption (instrumented by 91 SNPs) and serum iron/transferrin saturation were of a similar magnitude, CIs were wider. In sensitivity analyses solely using data from deCODE which did not adjust their GWAS for alcohol, overall findings were unchanged (**S8 Fig**). We had >80% power to detect a causal effect of 0.13 SD.

## Clinical relevance of elevated brain iron

Higher hippocampal and pallidum susceptibility were associated with slower TMT duration (right side, TMT A: β = $2.20 \times 10^{-2}$, 95% CI: $5.94 \times 10^{-3}$ to $3.80 \times 10^{-2}$, $p$ = 0.038) and lower

| Brain.IDP | SNPs | | Estimate | LCI | UCI | p.value |
|---|---|---|---|---|---|---|
| Left putamen | 92 | | 0.25 | 0.013 | 0.487 | 0.039 |
| Right putamen | 92 | | 0.208 | -0.041 | 0.458 | 0.102 |
| Left caudate | 92 | | -0.257 | -0.692 | 0.178 | 0.247 |
| Right caudate | 92 | | -0.257 | -0.688 | 0.174 | 0.243 |
| Left thalamus | 92 | | 0.142 | -0.063 | 0.347 | 0.174 |
| Right thalamus | 92 | | 0.131 | -0.081 | 0.342 | 0.225 |
| Left hippocampus | 92 | | 0.141 | -0.088 | 0.37 | 0.227 |
| Right hippocampus | 92 | | 0.276 | 0.053 | 0.498 | 0.015 |
| Left amygdala | 92 | | 0.095 | -0.118 | 0.308 | 0.384 |
| Right amygdala | 92 | | 0.04 | -0.166 | 0.245 | 0.705 |
| Left pallidum | 92 | | -0.367 | -0.853 | 0.119 | 0.139 |
| Right pallidum | 92 | | -0.436 | -0.926 | 0.054 | 0.081 |
| Left substantia nigra | 92 | | -0.311 | -0.759 | 0.138 | 0.174 |
| Right substantia nigra | 92 | | -0.325 | 0.312 | -0.963 | 0.171 |
| Left accumbens | 92 | | 0.041 | -0.204 | 0.287 | 0.741 |
| Right accumbens | 92 | | 0.02 | -0.213 | 0.253 | 0.867 |

-1.5  -1.0  -0.5  0.0  0.5  1.0

Lower iron ← → Higher iron

**Fig 4. Two-sample linear MR estimates for the causal effect of alcohol on susceptibility.** Genetic associations with alcohol consumption calculated from GWAS and Sequencing Consortium of Alcohol and Nicotine, associations with QSM image-derived phenotypes were derived from UKB. Effect estimates for alcohol consumption are per SD increase in genetically predicted log-transformed alcoholic drinks per week. Estimates from IVW analysis. IDP, quantitative susceptibility mapping imaging-derived phenotype; MR, mendelian randomization; QSM, quantitative susceptibility mapping; SNP, single nucleotide polymorphism; LCI, lower confidence interval; UCI, upper confidence interval; SD, standard deviation; UKB, UK Biobank.

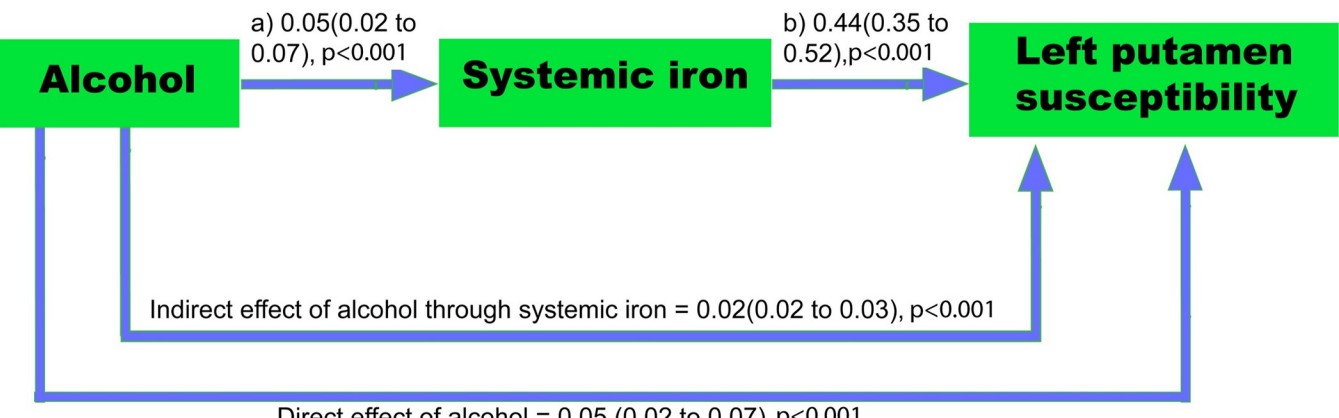

**Fig 5. Mediation of alcohol-related increases in left putamen susceptibility by liver iron.** *N* = 6,936 UKB participants. Numbers on the arrows are regression coefficients (with 95% CIs) reflecting: (a) change in liver iron (mg/g) for a 1 SD increase in alcohol intake weekly, (b) change in susceptibility (SD (95% CIs)) for a 1 mg/g increase in liver iron. The indirect and direct effects (standardized) derived from the mediation analysis are reported on the arrows linking alcohol to susceptibility. Models were adjusted for: age, sex, imaging site, Townsend Deprivation Index, educational qualifications, household income, historical job, blood pressure, BMI, cholesterol, smoking, and polygenic risk score for serum iron. BMI, body mass index; CI, confidence interval; SD, standard deviation; UKB, UK Biobank.

fluid intelligence (left: β = −2.98 × 10⁻², 95% CI: −4.59 × 10⁻² to −1.37 × 10⁻², *p* < 0.001) (**S10 Table**). Associations were bilateral for the pallidum, but associations with fluid intelligence stronger for the left hippocampus.

Additionally, there were significant interactions between age and bilateral putamen (TMT A: duration β = 0.006, 95% CI: 0.003 to 0.008, *p* < 0.001; TMT B: duration β = 0.004, 95% CI: 0.002 to 0.006, *p* < 0.001), bilateral caudate (TMT A: β = 2.87 × 10⁻³, 95% CI: 7.67 × 10⁻⁴ to 4.97 × 10⁻³, *p* < 0.001; TMT B: β = 3.37 × 10⁻³, 95% CI: 1.25 × 10⁻³ to 5.49 × 10⁻³, *p* < 0.001), and right hippocampal (TMT A: β = 2.95 × 10⁻³, 95% CI: 8.43 × 10⁻⁴ to 5.05 × 10⁻³, *p* < 0.001) susceptibility in models of executive function (**Fig 7A**–**7C**). Interactions with age were also

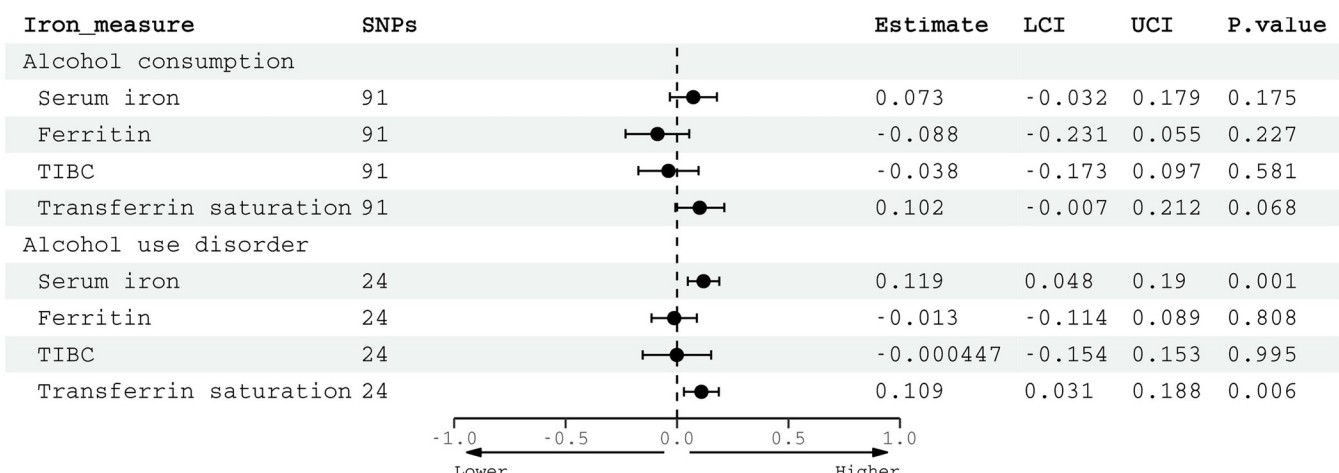

**Fig 6. Two-sample MR estimates of the causal effect of alcohol and alcohol use disorder on serum iron markers.** Genetic associations with: alcohol consumption calculated in GWAS and Sequencing Consortium of Alcohol and Nicotine, alcohol use disorder in Million Veterans Program and Psychiatric Genomics Consortium, and serum iron markers in deCODE, INTERVAL, and Danish Blood Donor Study. IVW estimates are shown. Effect estimates for alcohol consumption are per SD increase in genetically predicted log-transformed alcoholic drinkers per week, and for AUD having a diagnosis of AUD. AUD, alcohol use disorder; IVW, inverse variance weighted; LCI, lower confidence interval; MR, mendelian randomization; SD, standard deviation; SNP, single nucleotide polymorphisms; TIBC, total iron-binding capacity; UCI, upper confidence interval.

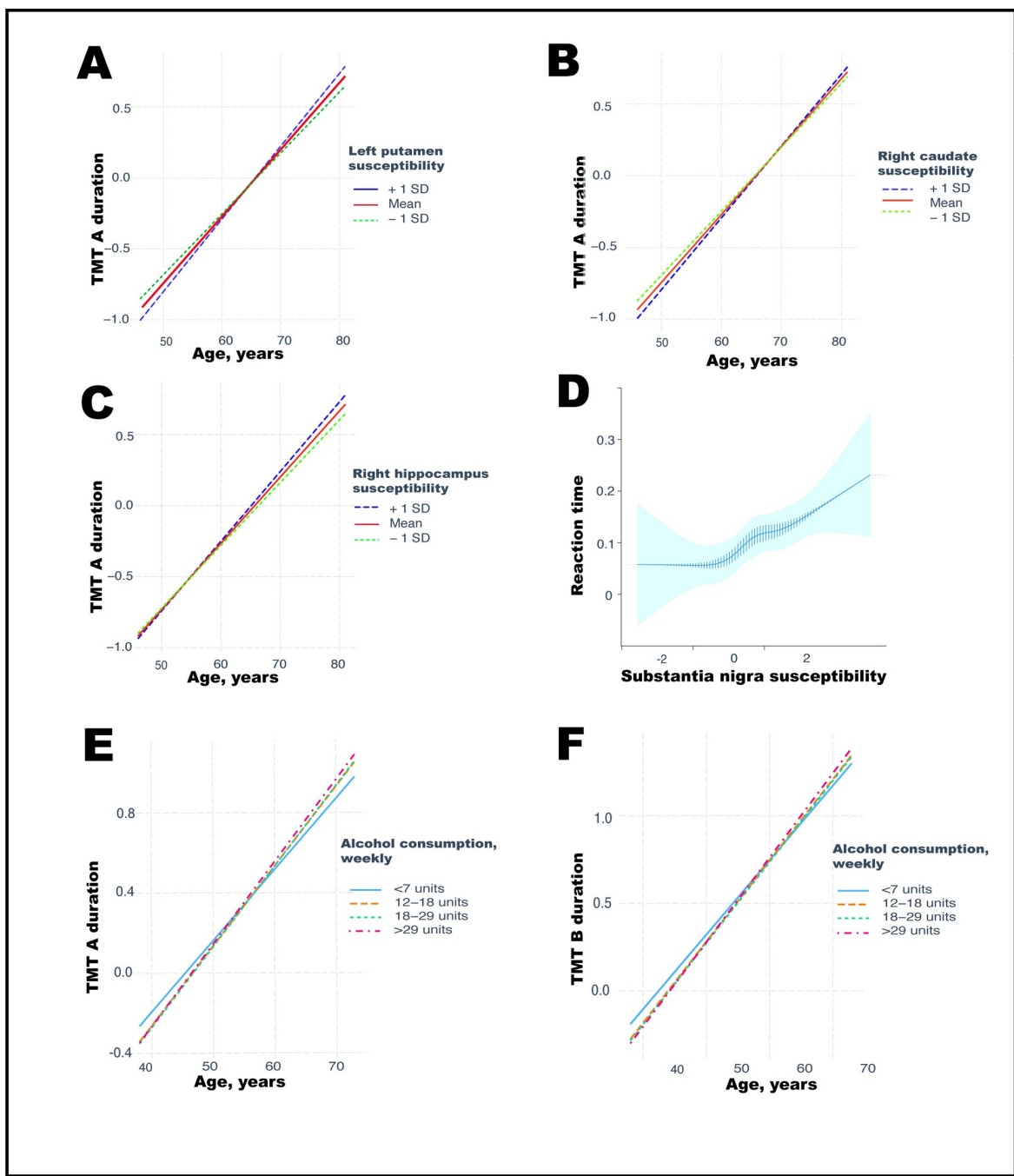

**Fig 7. Associations with cognitive function. (A)** Putamen susceptibility-dependent effect of age on executive function (at time of scan). TMT durations and susceptibility are quantile normalized. **(B)** Caudate susceptibility-dependent effect of age on executive function. **(C)** Hippocampal susceptibility-dependent effect of age on executive function. **(D)** Reaction time (quantile normalized and fitted with restricted cubic spline, 5 knots) according to substantia nigra susceptibility. **(E) and (F)** Alcohol consumption-dependent effect on executive function (at online follow up). Alcohol intake is weekly units categorised in quintiles. Only intakes significantly different from reference group (<7 units) are plotted for clearer visualisation. Graphs generated from regression models controlled for: age, sex, imaging site, diabetes, smoking, income, education, BMI, blood pressure, cholesterol, and historical job type. Abbreviation: SD–standard deviation. BMI, body mass index; SD, standard deviation; TMT, trail-making test.

evidence with putamen ($\beta$ = −0.004, 95% CI: −0.006 to −0.002, $p < 0.001$), caudate ($\beta$ = −$2.33 \times 10^{-3}$, 95% CI: −$4.47 \times 10^{-3}$ to −$1.93 \times 10^{-4}$, $p < 0.001$), and amygdala ($\beta$ = −$2.84 \times 10^{-3}$, 95% CI: −$4.97 \times 10^{-3}$ to −$7.04 \times 10^{-4}$, $p < 0.001$) susceptibility and fluid intelligence. In both cases, age interactions were stronger for the right brain regions than left.

Neither substantia nigra nor thalamus susceptibility was associated with cognitive function. However, higher right substantia nigra susceptibility ($\beta$ = $2.36 \times 10^{-2}$, 95% CI: $7.64 \times 10^{-3}$ to $3.95 \times 10^{-2}$, $p < 0.001$), in addition to bilateral pallidum (right: $\beta$ = $2.55 \times 10^{-2}$, 95% CI: $9.40 \times 10^{-3}$ to $4.15 \times 10^{-2}$, $p < 0.001$), was associated with slower simple reaction time although not with other motor deficits (**S10 Table**).

There were no significant associations between putamen, caudate, hippocampus, or thalamus susceptibility and simple reaction time at the time of scan, or with age interactions in reaction time (**S10 Table**). Similarly, there were no significant associations between susceptibility measures and self-reported walking pace (OR = 1.00, 95% CI: 0.97 to 1.03). Grip strength was positively associated with bilateral putamen and caudate susceptibility (**S10 Table**), with significant sex interactions. Males appeared to drive the associations.

In a post hoc analysis, squared alcohol consumption was associated with slower TMT performance (alcohol $\beta$ = −$4.09 \times 10^{-3}$, 95% CI: −0.02 to 0.01, $p = 0.53$; alcohol$^2$ $\beta$ = 0.01, 95% CI: $3.84 \times 10^{-3}$ to 0.018, $p = 0.003$), and there were significant age by alcohol interactions (in those drinking 12 to 18 versus <7 units weekly $\beta$ = −0.24, 95% CI: −0.40 to −0.09) (**Fig 7E and 7F**). However, this was only observed within the larger sample that participated in the later online follow up, preventing exploration of whether brain iron was mediating this pathway.

## Discussion

### Summary of findings

Alcohol consumption, including at low levels, was observationally associated with higher brain iron in multiple basal ganglia regions. There was some evidence supporting a causal relationship between genetically predicted alcohol consumption and putamen and hippocampus susceptibility, although this did not survive multiple testing correction. Alcohol was associated with both higher liver iron, an index of systemic iron load, and genetically predicted AUD associated with genetically predicted serum iron markers. Brain iron accumulation in drinkers was only partially mediated via higher systemic iron. Markers of higher brain iron (higher susceptibility) were associated with poorer executive function and fluid intelligence and slower reaction speed.

The accumulation of iron in the brain we observed in moderate drinkers overlaps with findings of an observational study in AUD. Higher putamen and caudate iron levels were described in a small study of males with AUD ($n$ = 20) [9]. These individuals were drinking substantially more than our sample—a mean of 22 standard drinks per day (>37 units daily). In contrast, we observed associations in those drinking just >7 units per week. A recent phenome-wide association study of quantitative susceptibility in the same dataset reported significant associations in basal ganglia regions with higher frequency binge drinking [10]. Regional heterogeneity in iron concentrations is well described although the aetiology is not understood [47]. The basal ganglia, including the putamen [48], have some of the highest iron concentrations in the brain and suffer the greatest age-related increases [49]. Interestingly, we found significant alcohol-age interactions with susceptibility, suggesting that alcohol may magnify age effects on brain iron. We are mindful however that within UKB, changes with age could represent a cohort effect. In this sample, associations with susceptibility and T2* measures were observed at lower alcohol intakes in females. In haemochromatosis, females are relatively protected against the clinical manifestations of iron overload through menstrual blood loss [50].

The majority of our included sample, however, (70%) was postmenopausal and menopause status did not alter alcohol–brain iron associations. Sex differences in alcohol metabolism therefore may be responsible. These findings do not support current UK "low risk" drinking guidelines that recommend identical amounts for males and females [51]. We found some support for a causal relationship between alcohol consumption and susceptibility in the putamen and hippocampus, and between AUD and putamen susceptibility in MR analysis. Although these associations did not survive multiple comparisons correction, they are in the same direction as the highly significant observational associations. Associations between genetically predicted alcohol and susceptibility in other regions were not significant. We suspect this results from our limited power to detect small associations despite the sample size, given that the genetic instruments explain less than 1% of the phenotypic variation in alcohol consumption [29]. Furthermore, weak instrument bias, in the direction of the null, may be contributing [52]. Using UKB for our calculations, about one third of SNPs we used to instrument alcohol consumption had F statistics <10 (**S11 Table**).

Our MR results provide evidence for a causal role of AUD in increasing serum iron and transferrin saturation, a sensitive marker of iron overload [53]. While genetically predicted alcohol use was not significantly associated with ferritin, this mirrors findings in early haemochromatosis, where ferritin levels can be normal and transferrin saturation is the earliest marker of iron overload [54]. The associations we found with liver iron, a reliable marker of systemic iron stores, were consistent with the serum results. In fact, in our study, which we believe the largest investigation of alcohol and liver iron by an order of magnitude [55], iron levels were the most sensitive liver marker of alcohol-related damage. Alcohol suppresses hepcidin production, the major hormone-regulating iron homeostasis [56]. This suppression increases intestinal absorption of dietary iron [57] and limits export of iron from hepatocytes. In our CMA, higher systemic iron levels only explained 32% of alcohol's effects on brain iron, suggesting other mechanisms are also involved. These could include an increase of blood–brain barrier permeability to iron, in turn mediated by reduced thiamine that commonly occurs in AUD due to a combination of inadequate dietary intake, reduced absorption, and metabolic changes [58,59]. In cerebral autosomal dominant arteriopathy with subcortical infarcts and leukoencephalopathy (CADASIL) patients, iron leakage has been linked to blood–brain barrier permeability [60]. Other possible mechanisms include dopamine surges following alcohol ingestion or chronic inflammatory processes [61]. The alternative possibility is that individuals with higher brain iron drink more alcohol. One potential mechanism for this is that tyrosine hydroxylase, an enzyme in the dopamine synthesis pathway, is iron dependent [62]. Dopamine has been linked to alcohol cravings in dependence [63]. For this reason, we used MR to support/refute the observational analyses.

Higher putamen and caudate susceptibility interacted with age in predicting executive function and fluid intelligence, but not with simple motor tasks. Most, but not all, previous work has highlighted the importance of the putamen to complex motor tasks [64]. Interestingly, both trail making and the fluid intelligence tasks were performed within a time limit, and perhaps represent a measure of motor response linked to cognition, rather than a simple motor response. TMTs appear to be among the most sensitive to aging effects in the UKB cognitive battery [34]. Frontal dysfunction is well described in chronic heavy alcohol use [65]. Several putamen metrics have been associated with executive function, including blood flow [66], structural atrophy [67], and functional connectivity [68]. Iron accumulation in the putamen has also been described in developmental stuttering [69] and CADASIL [70]. Although most studies of dietary iron and cognition have been in children or anaemic individuals, there is some evidence that high dietary iron associates with poorer cognition [71]. While sex differences in cognition have been described [72], it is difficult to disentangle differing iron levels

from hormonal factors in the aetiology. How iron deposition could result in cognitive deficits requires further investigation. Iron co-localises in the brain with tau and beta amyloid [73], and can cause apoptosis and ferroptosis [74]. Higher substantia nigra susceptibility associated with slower reaction speed. The substantia nigra plays a vital role in movement regulation, and iron deposition in the substantia nigra has been linked to Parkinson's disease [75,76], a disorder with marked impairments in reaction speed [77].

To our knowledge, this is the largest study of moderate alcohol consumption and multiorgan iron accumulation. It is also the first study to use MR to investigate causality of alcohol on serum and brain iron.

We did not observe widespread associations between susceptibility or $T2^*$ and other cognitive tests or self-reported motor measures. Brain iron is likely to be an early marker of disease, and participants may have been examined too early in the process to detect clinical manifestations. Additionally, we are not likely to have captured the best phenotypes to assess basal ganglia function in the absence of objective motor measurements such as gait speed or a pegboard test. Self-reported walking speed may poorly approximate actual motor function. The cognitive tests were limited in scope and concerns have been raised about the reliability of the tests used [34]. Healthy selection biases in UKB are well described, and are likely magnified in the imaging subsample, but will equally bias the study towards null results [78]. Furthermore, associations in UKB seem to track with those observed in representative cohorts [79].

Changes in $T2^*$ and $\chi$ can reflect changes in iron but also myelin [80,81]. One key difference between $T2^*$ and $\chi$ is that iron (paramagnetic) and myelin (diamagnetic) have the opposite effect on $\chi$ in QSM, but the same effect on $T2^*$. Hence, the positive associations we observed between $\chi$ and alcohol could theoretically be driven by increased iron or reduced myelin. If the latter, then alcohol would also be positively associated with $T2^*$ (reduced myelin leads to longer $T2^*$). In contrast, we observed negative associations between $T2^*$ and alcohol. This supports our interpretation that increased iron is driving our results, given one highly plausible assumption, that alcohol does not increase grey matter myelination [82,83].

Partial volume effects could confound associations between hippocampal susceptibility and alcohol. For example, hippocampal atrophy, previously observed in drinkers [1], could be conflated with changes in $\chi$. However, this would tend to reduce estimated $\chi$. Alcohol was self-reported, but this is the only feasible method to ascertain intake at scale. Serum markers of iron homeostasis were not directly measured in UKB. Although analyses were controlled for the strongest SNPs associated with serum iron, these are likely to explain a low proportion of the variance. MR techniques rely on a number of assumptions that we have tried to test where possible, but residual uncertainty inevitably remains. Estimates were calculated in European individuals, but it is unclear how they generalise to other populations. MR estimates the effect of lifelong exposure, which does not necessarily translate into potential effects resulting from an intervention in adult life. Liver $T2^*$ has been useful in some studies to monitor iron overload, but further validation of this biomarker as a diagnostic marker of iron overload is needed [84]. Genetic variants explain a low variance of alcohol traits. Therefore, our analysis within the imaging sample, despite its large size, has limited power to detect small effects. The power for the larger serum iron measures was greater. For this reason, although nonlinear relationships between alcohol and health outcomes are of interest, we limited MR analyses to linear models. Mediation analysis is not experimental in design, and relies on intervention–outcome, intervention–mediator, and mediator–outcome effects being unconfounded to permit valid causal inferences. Alcohol exposure prior to study baseline (left truncation) may bias observational estimates [85]. In this study, liver and brain iron were measured at the same time, meaning reverse causation is possible. However, it is difficult to conceive of a plausible mechanism by which brain iron levels could substantially affect systemic iron.

Never drinkers appeared to have the lowest levels of brain iron. This is in keeping with our earlier work indicating there may be no safe level of alcohol consumption for brain health [20]. Moderate drinking is highly prevalent, so if elevated brain iron is confirmed as a mechanism by which alcohol leads to cognitive decline, there are opportunities for intervention on a population scale. Iron chelation therapy is already being investigated for Alzheimer's and Parkinson's diseases [17,18,86]. Furthermore, if reduced thiamine is mediating brain iron accumulation, then interventions to improve nutrition and thiamine supplementation could be extended beyond harmful and dependent drinkers, as is currently recommended [87], to moderate drinkers.

## Conclusions

In this large sample of UKB participants, we find evidence for elevated susceptibility and reduced T2$^*$ in basal ganglia regions with even moderate alcohol consumption. These changes likely reflect increased iron concentrations. Alcohol-related brain iron may be partially mediated by higher systemic iron levels, but it is likely there are additional mechanisms involved. Poorer executive function and fluid intelligence and slower reaction speeds were seen with markers of higher basal ganglia iron. Brain iron accumulation is a possible mechanism for alcohol-related cognitive decline.

## Supporting information

**S1 Checklist. STROBE checklist of recommended items to address in cohort studies.**
(DOCX)

**S2 Checklist. STROBE-MR checklist of recommended items to address in reports of mendelian randomization studies.**
(DOCX)

**S1 Fig. Flow chart of participants included in analyses.** UKB, UK Biobank; QSM, quantitative susceptibility mapping.
(PNG)

**S2 Fig. Comparison of different 2-sample MR methods for estimating the causal effects of alcohol on brain susceptibility.** Genetic associations with alcohol consumption were calculated in GWAS and Sequencing Consortium of Alcohol and Nicotine, and genetic associations with susceptibility imaging-derived phenotypes from UKB. Effect estimates for alcohol consumption are per standard deviation increase in genetically predicted log-transformed drinks per week. LCI, lower confidence interval; MR, mendelian randomization; SNP, single nucleotide polymorphisms; UCI, upper confidence interval.
(PNG)

**S3 Fig. Two-sample MR randomization analyses to estimate the causal effects of alcohol use disorder on brain susceptibility.** MR estimates (2-sample design) for associations between genetically predicted alcohol use disorder (Million Veterans Program and Psychiatric Genomics Consortium) and susceptibility imaging-derived phenotypes (UK Biobank) in inverse-variance weighted analysis. LCI, lower confidence interval; MR, mendelian randomization; SNP, single nucleotide polymorphism; UCI, upper confidence interval.
(PNG)

**S4 Fig. Observational associations between weekly alcohol consumption (quintiles) and liver iron (mg/g).** Reference group is those drinking <7 units (56 g) weekly. Estimates generated from regression models adjusted for: age, educational qualifications, Townsend Deprivation Index, household income, historical job code, smoking, imaging site, diabetes mellitus,

BMI, blood pressure, cholesterol, dietary iron, rs1800562, rs1799945, and rs855791. BMI, body mass index; LCI, lower confidence interval; UCI, upper confidence interval.
(PNG)

**S5 Fig. Observational associations between weekly alcohol consumption (quintiles) and liver protein density fat fraction (%).** Estimates are adjusted for age, sex, smoking, BMI, cholesterol, blood pressure, diabetes, educational qualifications, Townsend Deprivation Index, household income, and historical job type. BMI, body mass index; LCI, lower confidence interval; UCI, upper confidence interval.
(PNG)

**S6 Fig. Observational associations between weekly alcohol consumption (quintiles) and liver cT1 (milliseconds), a marker of inflammation/fibrosis.** Estimates are adjusted for: age, sex, smoking, BMI, cholesterol, blood pressure, diabetes, educational qualifications, Townsend Deprivation Index, household income, and historical job type. All estimates are within the normal reference range [88]. BMI, body mass index; LCI, lower confidence interval; UCI, upper confidence interval.
(PNG)

**S7 Fig. Comparison of different two-sample MR estimates of the causal effect of alcohol used disorder on serum markers of iron homeostasis.** Genetic associations of alcohol use disorder generated in the Million Veterans Program and Psychiatric Genomics Consortium and of serum markers of iron homeostasis from deCODE, INTERVAL, and the Danish Blood Donor Study. LCI, lower confidence interval; MR, mendelian randomization; SNP, single nucleotide polymorphism; TIBC, total iron binding capacity; UCI, upper confidence interval.
(PNG)

**S8 Fig. Two-sample MR estimates of the causal effects of alcohol consumption and alcohol use disorder on serum iron markers.** Genetically predicted alcohol consumption was log-transformed drinks per week, generated from GWAS and Sequencing Consortium of Alcohol and Nicotine. Genetic associations with alcohol use disorder were generated from the Million Veterans Program and Psychiatric Genomics Consortium. Genetic associations with serum iron markers were calculated in cohorts that did not adjust for alcohol in their genome-wide association study (DECODE unless otherwise marked). LCI, lower confidence interval; SNP, single nucleotide polymorphism; UCI, upper confidence interval.
(PNG)

**S1 Table. Summary statistics sources for genetic associations with alcohol intake, alcohol use disorder, serum iron measures, and brain iron measures.**
(XLSX)

**S2 Table. Baseline characteristics according to alcohol intake.**
(XLSX)

**S3 Table. Observational associations between alcohol consumption and brain iron measures.**
(XLSX)

**S4 Table. Unadjusted observational associations between weekly alcohol intake (quintiles) and brain susceptibility.**
(XLSX)

**S5 Table. Observational associations between weekly alcohol intake (octiles) and brain susceptibility.**
(XLSX)

**S6 Table. Observational associations between (1) alcohol intake and brain susceptibility controlling for menopause status; and (2) susceptibility and menopause status.**
(XLSX)

**S7 Table. Baseline characteristics for sample with diet and iron supplementation data.**
(XLSX)

**S8 Table. Observational associations between weekly alcohol intake (quintiles) and brain susceptibility, additionally adjusted for diet and iron supplementation.**
(XLSX)

**S9 Table. Mediation analyses for all brain regions.**
(XLSX)

**S10 Table. Observational associations between brain susceptibility and cognitive test performance at time of the scan.**
(XLSX)

**S11 Table. F statistics for genetic instruments for alcohol consumption.**
(XLSX)

## Author Contributions

**Conceptualization:** Anya Topiwala, Klaus P. Ebmeier.

**Data curation:** Anya Topiwala, Chaoyue Wang, Steven Bell, Daniel F. Levey, Hang Zhou, Stephen M. Smith.

**Formal analysis:** Anya Topiwala, Chaoyue Wang, Hang Zhou, Joel Gelernter, Karla L. Miller.

**Funding acquisition:** Anya Topiwala.

**Investigation:** Anya Topiwala, Daniel F. Levey.

**Methodology:** Anya Topiwala, Chaoyue Wang, Stephen Burgess, Steven Bell, Daniel F. Levey, Adriana Roca-Fernández, Steffen E. Petersen, Betty Raman, Masud Husain, Joel Gelernter, Karla L. Miller, Stephen M. Smith, Thomas E. Nichols.

**Project administration:** Anya Topiwala, Celeste McCracken.

**Software:** Adriana Roca-Fernández.

**Supervision:** Klaus P. Ebmeier, Stephen Burgess, Daniel F. Levey, Masud Husain, Joel Gelernter, Karla L. Miller, Thomas E. Nichols.

**Visualization:** Anya Topiwala.

**Writing – original draft:** Anya Topiwala.

**Writing – review & editing:** Anya Topiwala, Chaoyue Wang, Klaus P. Ebmeier, Stephen Burgess, Steven Bell, Daniel F. Levey, Hang Zhou, Celeste McCracken, Adriana Roca-Fernández, Steffen E. Petersen, Betty Raman, Masud Husain, Joel Gelernter, Karla L. Miller, Stephen M. Smith, Thomas E. Nichols.

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
