## [Editor Report · Decision Letter 0]

12 Feb 2022

Dear Dr Topiwala, 

Thank you for submitting your manuscript entitled "Impact of moderate alcohol consumption on brain iron and cognition: observational and genetic analyses" for consideration by PLOS Medicine.

Your manuscript has now been evaluated by the PLOS Medicine editorial staff and I am writing to let you know that we would like to send your submission out for external peer review.

Please re-submit your manuscript within two working days, i.e. by Feb 16 2022 11:59PM.

Kind regards,

Caitlin Moyer, Ph.D.

Associate Editor

PLOS Medicine

---

## [Decision Letter · Decision Letter 1]

23 Mar 2022

Dear Dr. Topiwala,

Thank you very much for submitting your manuscript "Impact of moderate alcohol consumption on brain iron and cognition: observational and genetic analyses" (PMEDICINE-D-22-00417R1) for consideration at PLOS Medicine. 

Your paper was evaluated by a senior editor and discussed among all the editors here. It was also discussed with an academic editor with relevant expertise, and sent to four independent reviewers, including a statistical reviewer. The reviews are appended at the bottom of this email and any accompanying reviewer attachments can be seen via the link below:

[LINK]

In light of these reviews, I am afraid that we will not be able to accept the manuscript for publication in the journal in its current form, but we would like to consider a revised version that addresses the reviewers' and editors' comments. Obviously we cannot make any decision about publication until we have seen the revised manuscript and your response, and we plan to seek re-review by one or more of the reviewers. 

We expect to receive your revised manuscript by Apr 13 2022 11:59PM. Please email us (plosmedicine@plos.org) if you have any questions or concerns.

We look forward to receiving your revised manuscript. 

Sincerely,

Caitlin Moyer, Ph.D.

Associate Editor

PLOS Medicine

plosmedicine.org

1. Competing Interests: Please add this statement to the manuscript's Competing Interests: "SB is a paid statistical consultant on PLOS Medicine's statistical board."

2. Data Availability Statement: The Data Availability Statement (DAS) requires revision. For each data source used in your study:

Please also indicate how any scripts necessary for analysis may be accessed.

3. Throughout: Please include line numbers running throughout the text.

4. Abstract: Please structure your abstract using the PLOS Medicine headings (Background, Methods and Findings, Conclusions).

5. Abstract: Methods and Findings: Please mention the years during which the study took place, when imaging took place relative to baseline assessments, length of follow up for the cognition assessment, and details of main outcome measures.

6. Abstract: Methods and Findings: Please quantify the main results presented with both 95% CIs and p values.

7. Abstract: Methods and Findings: In the last sentence of the Abstract Methods and Findings section, please describe the main limitation(s) of the study's methodology.

8. Abstract: Conclusions: Here, and throughout, please temper assertions such as “This study represents the largest…” with “To the best of our knowledge…” or similar. Please interpret the study based on the results presented in the abstract, emphasizing what is new without overstating your conclusions.

9. Author summary: At this stage, we ask that you include a short, non-technical Author Summary of your research to make findings accessible to a wide audience that includes both scientists and non-scientists. The Author Summary should immediately follow the Abstract in your revised manuscript. This text is subject to editorial change and should be distinct from the scientific abstract. Please see our author guidelines for more information: https://journals.plos.org/plosmedicine/s/revising-your-manuscript#loc-author-summary

10. Introduction: Please avoid vague statements such as “...there are huge public health implications.”

11. Introduction: Please conclude the final paragraph of the Introduction with a clear description of the study’s objectives (in paragraph form, rather than as a numbered list).

12. Methods: Participants: Please clarify if participants with dementia at baseline were excluded. Please provide more detail on how those participants invited for brain/abdominal imaging were selected.

13. Methods: Alcohol consumption: Please clarify that only the current drinkers were categorized into quintiles (excluding former/never drinkers).

14. Methods: Brain imaging: “The fourteen regions correspond to left and right of seven subcortical structures.” Please list the regions, with rationale for why they were selected. Please clarify if data from left/right substantia nigra were available only for QSM (as it seems there were 16 total QSM IDPs).

15. Methods: Genetic variants: For each data source, please describe the study design, size, and the underlying population. Please provide relevant details on selection of the genetic variants. Please consider including a supporting information table to describe sources of data.

16. Methods: Clinical measures: Please clarify in the text which elements of the cognitive battery were assessed at baseline and at follow up. Please clarify the time point of the follow up assessment. Please describe and reference how prospective memory and fluid intelligence were assessed. Please also clarify how the subset of 100,000 participants were invited for the online follow up trail making test assessment.

17. Statistical analyses: Did your study have a prospective protocol or analysis plan? Please state this (either way) early in the Methods section.

18. Statistical analyses: Observational: Please clarify how each covariate was assessed, categorized, and incorporated into analyses.

19. Statistical analyses: Observational: Please clarify the numbers of participants with diet/supplement data available, and whether this was missing from other participants, or if this was evaluated in only an invited subset (and how this subset was selected): “For the subset with available data, dietary factors (meat, fish, bread, fruit and vegetable consumption) and dietary iron supplementation were also included as covariates, to test whether differences in diets according to alcohol intake were driving associations.” Please also clarify how diet and supplement data were assessed in participants, and ensure that the results includes the number of participants for which these measures were available in a Table summarizing the characteristics of this sub-sample.

20. Statistical analyses: Observational: Please clarify if participants were excluded based on having an iron chelation prescription: “No included subjects reported iron chelator prescription.”

21. Statistical analyses: Observational: Please clarify how the repeated assessment of TMT (at baseline and follow up) was incorporated into the analysis.

22. Statistical analyses: Causal mediation analysis: Please provide additional detail of the CMA. Please specify, the details of the regression model, the inclusion of variables, confounders, and interaction terms, assumptions, and how missing data were dealt with.

23. Statistical analyses: Mendelian randomization: Please provide complete details for the Mendelian randomization methods used and report the study according to STROBE-MR guidelines. Please mention the three core IV assumptions for the main analysis (relevance, independence and exclusion restriction). Please mention how missing data were dealt with.

24. Statistical analyses: Mendelian randomization: Please comment on risk of bias attributable to participant overlap between the datasets used.

25. Reporting: Please ensure that the observational component of the study is reported according to the STROBE guideline, and include the completed STROBE checklist as Supporting Information. Please ensure that the Mendelian randomization portion of the study is reported according to the STROBE-MR guideline, and include the completed STROBE-MR checklist as Supporting Information.

Please add the following statement, or similar, to the Methods: "This study is reported as per the Strengthening the Reporting of Observational Studies in Epidemiology (STROBE) guideline, including guidelines specific for Mendelian randomization studies (S1 and S2 Checklist)." or similar.

The STROBE-MR guideline can be found here: https://www.strobe-mr.org/

When completing the checklists, please use section and paragraph numbers, rather than page numbers.

26. Results: Please report the complete results from the study, including the results for each brain region. Currently, results are presented for individual brain regions selectively. PLOS does not permit “data not shown” and request that results for each brain region be provided (at least in the Supporting Information).

27. Results: For all observational analyses, please present the results of both unadjusted and adjusted analyses. Please consistently present both 95% CIs and p values in the text and tables.

28. Results: Associations with brain iron: Please remove the duplicate text here “0.09 standard deviation (S.D.) [95% confidence interval 0.07 to 0.10]), caudate (β=0.06 [0.04 to 0.07]) and substantia nigra (β=0.04 [0.02 to 0.05]), and lower iron in the thalami (β= -0.05 [-0.06 to -0.03” and please report p values for these associations.

29. Results: It is not clear why the results from left putamen are being highlighted. Please provide similar results (e.g. Figure 2A) for all brain regions.

30. Results: “A sensitivity analysis with finer categorisation of alcohol intake confirmed that associations were not observed at lower intakes (SFigure 1).” Please clarify if this means intakes lower than 7 units.

31. Results: “Menopause status did not associate with putamen susceptibility in females, nor did it not alter associations between alcohol and susceptibility (STable 1).” Please clarify if associations by menopause status was investigated for all brain regions.

32. Results: “In the subset with complete data, neither adjustment for dietary factors, including frequency of red meat consumption (SFigure 8), nor dietary iron supplementation (SFigure 9) changed the pattern of associations.” Please provide a comparison of characteristics for those with compared to without these data available (including information on diet and supplementation).

33. Results: Pathways from alcohol to brain iron: Please clarify if the mediation analysis was carried out for each brain region. It is not clear if the example results presented are from the left or right putamen. Please provide the results of the mediation analysis for each brain region in the study in the supporting information files.

34. Results: “Alcohol consumption (>11 units) was associated with higher liver iron measured by MRI, a robust marker of systemic iron stores” Please provide the results supporting this finding, with 95% CIs and p values.

35. Results: “Higher putamen, caudate and hippocampal susceptibility were associated with greater age-related slowing of executive function, and higher putamen and caudate susceptibility with greater age-related differences in fluid intelligence (Figure 6 (A-C) & STable 3).” Age-related slowing seems to imply you are referring to the repeated assessment of the Trail Making Test, while the figures seem to describe cross-sectional associations. Please revise to clarify. Please also clarify why results for putatmen, caudate, and hippocampus are mentioned- and whether findings differed bilaterally (i.e. the choice to include left vs. right in Figure 6).

36. Results: Clinical relevance of elevated brain iron: Please report p values for all results reported, alongside the 95% CIs. Please explain when findings for left vs right hemisphere brain regions are described, whether findings are consistent bilaterally.

37. Discussion: Please present and organize the Discussion as follows: a short, clear summary of the article's findings; what the study adds to existing research and where and why the results may differ from previous research; strengths and limitations of the study; implications and next steps for research, clinical practice, and/or public policy; one-paragraph conclusion.

38. Discussion: Key findings: Please differentiate between observational evidence, and evidence derived from genetically predicted alcohol use. Please describe as “evidence supporting a causal relationship between AUD and higher serum iron” or similar.

39. References: Please use the "Vancouver" style for reference formatting, and see our website for other reference guidelines https://journals.plos.org/plosmedicine/s/submission-guidelines#loc-references:

40. Table 1: Please define units of alcohol in the legend. Please also describe “higher degree” in the legend. In the Methods, please describe how education level was categorized. Please report on all relevant relevant variables incorporated into the analyses.

41. Figure 2: Please include a larger version of Panel A, rather than including this as an inset. Please include p values for Panel A. For the graph, it would be helpful to include more descriptive terms with the axis labels. Please also explain the color code system for the markers.

42. Figure 3: Please also report p values.

43. Figure 5: Please also report p values.

44. Supporting Information: Please make sure that each Supporting Information Figure/Table has a title and a legend.

45. Figures S1 - S9: Please provide p values. Please provide the results of sensitivity analyses for all brain regions included in the study (by octiles of alcohol consumption, with adjustments for diet and iron supplementation).

46. Figure S10: Please move the inset (A) to below the graph, and increase the size. Please include p values with the 95% CIs reported in the panel. The rationale for illustrating findings from right putamen are not clear. Please provide the result for the T2* derived analyses for each brain region in the study, in addition to the right putamen, especially given that the left putamen was selected for illustration in Figure 2.

47. Figure S11: Please include p values and please include the results for each brain region included in the study.

48. Figure S12, S13, S14, S15, S16: Please include p values.

49. Table S1: Please report exact p values, please do not report p<0.05. Please include the data for each brain region in the study.

50. Table S2: It may be helpful to present these data similar to Figure 3 in the main text. Please report exact p values rather than p<0.05.

51. Table S3 and Table S4: Please report exact p values rather than p<0.05, etc. Please present associations for all brain regions included in the study.

Comments from the reviewers:

Reviewer #1: This is an interesting paper testing the somewhat novel hypothesis that alcohol causes cognitive decline via raising iron, using an observational and Mendelian randomization study.

Methods

To clarify the observational study please address the following points

1. Please ensure that the analysis is adjusted comprehensively for confounders. Confounders are typically common causes of exposure and outcome, as explained here. Socio-economic position is often an influential confounder that likely affects many estimates not just those concerning education

2. Please ensure the analysis is not adjusted for consequences of the exposure unless they are independent of the outcome. Adjusting for non-confounders can bias estimates as explained here https://pubmed.ncbi.nlm.nih.gov/16931543/. 

3. When assessing interactions, please ensure interactions with confounders are included in the model, as explained here https://pubmed.ncbi.nlm.nih.gov/22422832/

4. Please explain how the issue of the exposure starting before recruitment may affect the estimates as explained here https://pubmed.ncbi.nlm.nih.gov/27237061/

5. Please provide tables showing exposures by potential confounders so that the reader can assess the likely level of confounding

6. Please also provide a table showing how the sample with brain and liver imaging relates to UK Biobank on key characteristics, so that the reader can assess any issues arising from selection

As regards the Mendelian randomization (MR) study, please clarify the following points in the manuscript

1. Were the SNPs used as genetic instruments independent?

2. Were the SNPs used as genetic instruments for iron independent of hepcidin and hereditary hemochromatosis? 

3. Did the inverse variance weighted estimates use multiplicative effects?

4. Please include power calculations

5. Would it be possible to use multivariable MR to show that the effect of alcohol on cognition is mediated by iron? 

Discussion

To contextualize the results please explain whether any other drivers of iron levels, such as supplements or foods rich in iron, would be expected to, or are known to, have detrimental effects on cognition. Similarly, iron levels vary by sex and population, are corresponding differences in cognition seen? 

In the interpretation, please consider whether the changes with age could be a cohort effect. The UK Biobank participants encompass a wide age range recruited over a short time span, so age also represents cohort.

Please give more thorough consideration of the limitations of the methods used.

Reviewer #2: In this new manuscript, Topiwala et al report an association between alcohol consumption and MRI-determined brain iron. This study has public health importance, since the association was observed in even moderate drinkers. The importance of brain iron to neurological diseases is increasingly appreciated, and a proposed model of iron elevation mediating alcohol-related cognitive deficits is plausible. The study examined a large and well characterized dataset (n>22k) of brain MR images, and combined this with additional genetic data, clinical characteristics (including several risk factors), cognitive outcomes, and systemic values of iron from plasma and liver MRI. The authors note a major, but almost unavoidable limitation of this dataset is that alcohol use is self-reported, but otherwise this represents an ideal dataset to perform this study on. This limitation is somewhat mitigated by the use of genetic predictors of alcohol use disorder, where the authors report that these too are associated with brain iron in certain regions. Overall, I found this to be a well conducted, comprehensive, and important study. I have the following comments:

* Age and sex are likely to influence brain iron. I note that the authors controlled for these variables, but it is perhaps likely that the association between alcohol use and brain iron may be non-linear with age, or may be more apparent in either males or females. Is an interaction term of age*alcohol or sex*alcohol significant? Or are there different results when comparing associations in strata of sex or age (young/old). It may be of public health importance to know whether, for example, the association only occurs after a certain age, or only in males. 

* The figures were distorted when printed to the PDF. There appear to be random boxes covering parts of the figures, and other significant graphical issues. 

* In Figure 2, the colours of the dots are not explained. The inset is hard to read and should be included as a separate table. 

* The authors make the case for alcohol use causing brain iron elevation. I think this is likely, but would the authors consider also the reverse possibility - that people with high brain iron are more likely to drink alcohol. A possible mechanism for this is that tyrosine hydroxylase (dopamine-producing enzyme) is dependent on iron. 

Scott Ayton

Reviewer #3: The authors use observational and Mendelian randomization (MR) analyses to investigate whether moderate alcohol consumption leads to higher levels of iron concentration in various brain regions, whether this effect of alcohol on brain iron is mediated by systemic iron levels, and whether higher concentration of brain iron causes a sharper decline in age-related cognitive function.

The manuscript uses suitable large-scale datasets, makes use of appropriate methodology, links the obtained results with existing literature and discusses the biological plausibility and clinical relevance of novel findings. I enjoyed reading the manuscript and believe it is already deserving of publication. I have only a few minor comments:

1) The effects of alcohol consumption on health-related traits are sometimes non-linear. At the same time, the authors state that part of their motivation for conducting this study is to consider the impact of moderate alcohol consumption on brain iron levels, as opposed to heavy consumption which has already been investigated in the literature. With that in mind, did the authors observe any evidence of non-linear behaviour in the effects of alcohol consumption on brain iron levels and cognitive function?

2) The authors mention that the genetic variants used in their study explained less than 1% of variation in alcohol consumption. Does that mean their MR analysis could be susceptible to weak instrument bias? Since the paper uses a two-sample design, any such bias would act towards the null, making it harder to identify significant associations between drinking and iron levels. Perhaps it would be a good idea to report F-statistics as an indication of instrument strength in the authors' MR analyses.

3) There must be an issue with the confidence intervals plotted in Figure 6, as the "+1 SD" and "-1 SD" lines seem to be reversed for smaller ages.

4) Directly below Figure 2, a few lines of text have been duplicated - please delete.

Reviewer #4: This is a comprehensive, well-conducted and clearly written examination of the relationship between brain iron and alcohol use. The study includes robust methods and appropriate sensitivity analyses, leading to convincing evidence that low levels of alcohol use are associated with iron accumulation in the brain. Given the prevalence of low-level alcohol use, the findings have considerable public health implications, particularly in terms of improving cognitive health. There are some details we feel should be addressed:

1) The models predicting alcohol use did not include sociodemographic covariates which were included in the models with cognitive outcomes. These covariates seem relevant to the alcohol analyses as well and we wondered why they weren't included? It was also unclear why the models investigating the relationship with serum iron (figures S13 and 14) then included the sociodemographic factors?

2) What was the motivation for including the analysis of genetically predicted AUD? Much of the introduction focuses on moderate levels of alcohol use and the importance of understanding the impact of this from a public health perspective so the inclusion of AUD is surprising. In addition, the statistical analysis section focusing on MR only mentions alcohol consumption not AUD.

3) When describing figures S2-6 the authors state: "Levels of drinking necessary to observe higher susceptibility values differed according to region". It may be worth noting in the text that drinking 7+ units was associated with higher susceptibility for all regions.

4) For sensitivity analyses which stratified the sample by sex and included diet-related covariates, only results for left putamen are shown. This is fine, but please confirm in text that only this is shown for illustrative purposes and that the analyses were conducted for all IDPs and that the pattern of relationships was similar across IDPs (if this is the case). Please also provide n for the sample including diet-related factors.

5) In terms of categorisation of alcohol consumption, the categories overlap slightly (i.e., 7-12, 12-18 etc). Please confirm this is a labelling error and correct if this is the case. The analyses for QSM and T2* seem to have alcohol categorised slightly differently - is there a reason for this? 

6) What is the sample size for the serum iron analyses?

7) Can the authors please provide more detail on their causal mediation analysis and how what is presented is different to standard mediation analysis.

8) Given the sample size, one-sample non-linear MR would not have been possible. But given that many studies do find a non-linear relationship between alcohol use and health outcomes (specifically a protective effect of low consumption) consideration of this as a limitation may be included.

Overall, we would like to commend the authors on an impressive piece of work.

Dr Louise Mewton & Ms Rachel Visontay

Centre for Healthy Brain Ageing, University of New South Wales, Sydney, Australia

[LINK]

---

## [Decision Letter · Decision Letter 2]

23 May 2022

Dear Dr. Topiwala,

Thank you very much for re-submitting your manuscript "Impact of moderate alcohol consumption on brain iron and cognition: observational and genetic analyses" (PMEDICINE-D-22-00417R2) for review by PLOS Medicine.

I have discussed the paper with my colleagues and the academic editor and it was also seen again by four reviewers. I am pleased to say that provided the remaining editorial and production issues are dealt with we are planning to accept the paper for publication in the journal.

[LINK]

We look forward to receiving the revised manuscript by May 30 2022 11:59PM.   

Sincerely,

Caitlin Moyer, Ph.D.

Associate Editor 

PLOS Medicine

plosmedicine.org

Requests from Editors:

1. Title: We suggest revising to: “Associations between moderate alcohol consumption, brain iron, and cognition in UK Biobank participants: Observational and Mendelian randomization analyses” or similar.

2. Data Availability statement: Your data availability statement currently reads: “Data access on successful UKB data application” in your data availability statement. The Data Availability Statement (DAS) requires revision. We ask that you please note the sources and location, contact information for data access requests, and any relevant DOI or accession number associated with the dataset(s) used in the study. 

In the response to editor/reviewer comments letter, you wrote: “Imaging and observational data underlying the results presented are available from the UK

Biobank upon successful application (https://www.ukbiobank.ac.uk/enable-yourresearch/apply-for-access). Genetic summary statistics for serum iron measures are freely available (https://www.decode.com/summarydata/), as are GSCAN summary statistics (https://genome.psych.umn.edu/index.php/GSCAN). Summary statistics for alcohol use disorder are available upon application through dfGaP at accession no. phs0016732.v3.p1 (https://www.ncbi.nlm.nih.gov/projects/gap/cgi-bin/study.cgi study_id=phs001672.v3.p1). Requests for code can be made to the authors.”

Please add this statement to the data availability section of the manuscript submission system. 

Regarding the sentence about the access to the analysis code, please note that authors may not serve as the contact points for access requests. All data and related metadata underlying reported findings should be deposited in appropriate public data repositories, unless already provided as part of the submitted article (i.e. as a supporting information file). When possible, we recommend authors deposit restricted data or code to a repository that allows for controlled data access. If this is not possible, directing data requests to a non-author institutional point of contact, such as a data access or ethics committee, helps guarantee long term stability and availability of data. Providing interested researchers with a durable point of contact ensures data will be accessible even if an author changes email addresses, institutions, or becomes unavailable to answer requests.

Please see our policy at http://journals.plos.org/plosmedicine/s/data-availability

3. Abstract: Please combine the Methods and Findings sections into one section, “Methods and findings”.

4. Abstract:Background: Line 95-96: We suggest revising to: “Our objectives were to investigate evidence in support of causal relationships between…”

5. Abstract: Methods and Findings Line 102:: We suggest moving “Observational associations between brain iron markers and self-reported alcohol consumption (n=20,729 UK Biobank participants) were compared with associations with genetically-predicted alcohol intake from two-sample Mendelian randomization (MR).” to the first sentence of the Methods and Findings section. Please indicate the years during which the alcohol consumption data were collected, if different from the study baseline years (2006-2010). Please report the percentage of female participants, and other relevant demographics or inclusion/exclusion criteria.

6. Abstract: Methods and Findings: Line 103-105: Please provide a few words explaining what is represented by each measure (QSM, susceptibility (χ) and T2*) for a non-specialist audience. Please mention the brain regions of interest in the study, at least those for which findings are presented, or mention specific pathways of interest (e.g. subcortical regions involved in motor behavior/learning/cognition, memory, etc).

7. Abstract: Methods and Findings: Please mention how alcohol consumption was evaluated, and please also note that genetically predicted alcohol use disorder was included in the MR anaysis. 

8. Abstract: Methods and Findings: Line 111: Please note the measures used to assess cognition, in terms of how the outcomes of executive function, fluid intelligence, reaction time were assessed.

9. Abstract: Methods and Findings: Please provide some summary demographics of the sample, e.g. sex, age, weekly alcohol consumption, proportion of never drinkers.

10. Abstract: Methods and Findings: Please report p values as p<0.001 where applicable.

11. Abstract: Methods and Findings: Please revise to indicate that the MR evidence did not survive correction for multiple testing: “MR analyses provided weak evidence that these relationships are causal. A 1 S.D. higher genetically-predicted number of alcoholic drinks weekly associated with 0.25 S.D. (95% CI:0.01 to 0.49, p=0.04) higher putamen susceptibility and 0.28 S.D. (95% CI:0.05 to 0.50, p=0.02) higher hippocampus susceptibility; however these associations did not survive corrections for multiple testing. Weak evidence for a causal relationship between genetically predicted alcohol use disorder and higher putamen susceptibility was observed (0.18, 95% CI:0.001 to 0.35, p=0.04 S.D.), however, this was not robust to correction for multiple comparisons.” or similar wording.

12. Author Summary: Line 175: We suggest revising to: “These findings suggest that moderate alcohol consumption is associated with higher iron levels in the brain.”

13. Author summary: Line 179: Please provide a few additional words of detail on the limitation of generalizability in your study.

14. Introduction: Line 186: Here and throughout, please remove spaces from reference brackets [1,2].

15. Introduction: Line 216: Please temper with “To the best of our knowledge…” or similar.

16. Introduction: Lines 220 - 228: This description of the various MRI-derived measurements may be more appropriate in the Methods section (e.g. around line 282 “Brain Imaging”).

17. Methods: LIne 252: Please explain how subsets were selected for imaging: “Subsets of these participants (all ancestry) were invited to have brain (n~50,000 analysed to date) and abdominal imaging (n~15,000 analysed to date).”

18. Methods: Line 315: Please also explain how “some participants” came to be invited to undergo liver imaging: “During the same visit that brain imaging was performed, some participants also underwent abdominal imaging…”

19. Methods: Line 331: Genetic variants: Please reference the supporting information file containing information on the 91/92 SNPs for genetically predicted alcohol consumption, and the 24 variants used for the alcohol use disorder analysis.

20. Methods: Line 361: Clinical measures: Please provide additional detail of the outcomes of the trail-making test; prospective memory shape taks, fluid intelligence measures, and motor function tasks (e.g. please include the assessments/questions and any scoring criteria as supporting information documents, or describe in detail in the Methods section).

21. Methods: Line 507: Please reference the website in the reference list. Please refer to the reference guidelines for details: https://journals.plos.org/plosmedicine/s/submission-guidelines#loc-references

22. Results: Please report p values as p<0.001 where applicable.

23. Results: Line 566: If relevant, it may be useful to note here that for the finer categorization analysis, the reference category was fewer than 4 units weekly.

24. Results: Line 567-568: Please clarify if this should be: “slightly higher susceptibility in left thalamus…

25. Results: Line 575: Please clarify if this should be: “There were significant interactions with age in bilateral putamen and caudate…

26. Results: Line 580-582: Please clarify this sentence: “Findings with T2* were broadly consistent, although in more limited brain regions (putamen and caudate solely where correlation between T2* and susceptibility higher r= -0.58-0.75).” Please make it clear that this is saying that for T2* associations were only observed in putamen and caudate, and that these are two regions where T2* and susceptibility measures are more highly correlated (if this is accurate).

27. Results: Line 759: The analysis with squared alcohol consumption was not described in the Methods. Please describe this analysis, including the rationale for examining squared consumption with cognitive measures.

28. Discussion: Please check the organization of the Discussion, At Line 877-888: Please make the sections of the Discussion describing strengths and limitations of the study more apparent. We suggest describing the study strengths in one or more paragraphs, followed by a discussion of the limitations in one or more paragraphs.

29. Discussion: Line 934: Please avoid the use of vague statements such as “There are major public health implications…” and instead please provide greater description/discussion of such implications.

30. Discussion: Line 952: We suggest revising to: “Brain iron accumulation is a possible mechanism for alcohol-related cognitive decline.”

31. Line 957: Conflict of Interest: Please remove the conflict of interest statement from the main text. Please be sure all information is entered into the “Competing Interests” section of the manuscript submission system.

32. Line 964: Financial disclosure: Please remove this section from the main text, and please be sure all information is completely and accurately entered into the “Funding” section of the manuscript submission system.

33. Line 996: Data availability: Please remove this section from the main text, and please be sure all information is completely and accurately entered into the “Data availability” section of the manuscript submission system.

34. References: Please check the formatting of each reference. Please use NLM journal title abbreviations. Please update any preprints, where applicable (e.g. Ref 38). Please use the "Vancouver" style for reference formatting, and see our website for other reference guidelines https://journals.plos.org/plosmedicine/s/submission-guidelines#loc-references

35. Figures and Tables: Please check each figure/table and ensure that all abbreviations used are defined in each legend, or within the table/figure itself.

36. Figure 2, 3: Please also report the unadjusted associations (this may be in a separate supporting information table). Please report p values as p<0.001 where applicable. Please report p values to 2 decimal places, or 3 decimal places if p<0.01.

37. Figure 5: Please report p values as p<0.001 where applicable. Please report p values to 2 decimal places, or 3 decimal places if p<0.01.

38. S4 Figure, S5 Figure, S6 Figure: Please use “ref” or similar for the <7 units line instead of indicating “0” and please report p values as p<0.001 where applicable.

39. STables: Please also provide a list of Titles/Legends for the two Checklists, and the Supporting Information Tables.

40. S Table 3 and S Table 4: Please provide p values as p<0.001 where applicable. Please report to two decimal places, or 3 decimal places if p<0.01. Please also provide unadjusted associations.

41. S Table 5, S Table 8: Please provide p values as p<0.001 where applicable. Please report to two decimal places, or 3 decimal places if p<0.01.

42. S Table 9: Please provide p values as p<0.001 where applicable. Please report to two decimal places, or 3 decimal places if p<0.01. Please also provide unadjusted associations.

43. S Table 10: Please provide these data for the alcohol use disorder analysis.

Comments from Reviewers:

Reviewer #1: Thank you for this revision I have no further comments

Reviewer #3: I am happy with the authors' response to the comments previously made by me and other reviewers. Therefore, I would like to recommend that their manuscript is accepted for publication.

Reviewer #4: Thank you for attending to our comments. No further comments.

[LINK]

---

## [Editor Report · Decision Letter 3]

1 Jun 2022

Dear Dr Topiwala, 

On behalf of my colleagues and the Academic Editor, Perminder Singh Sachdev, I am pleased to inform you that we have agreed to publish your manuscript "Associations between moderate alcohol consumption, brain iron, and cognition in UK Biobank participants: Observational and Mendelian randomization analyses" (PMEDICINE-D-22-00417R3) in PLOS Medicine.

Please also address the following editorial requests:

-Abstract: Line 126: Please change “found these effects” to “found these associations”

-Abstract: Line 139: Please change to “alcohol use was self-reported”

-Author summary: Line 169: Please change to “cognitive measures associated with higher brain iron” or similar.

-Author summary: Line 177: Please change to “suggest that moderate alcohol consumption is associated with higher iron levels in the brain”

-Table 1: Please add the mean/SD for BMI for the larger UK Biobank sample.

-References: Please check and correct all journal title abbreviations to the appropriate format (e.g. in Ref. 1 bmj should be BMJ).

PRESS

Sincerely, 

Caitlin Moyer, Ph.D. 

Associate Editor 

PLOS Medicine